# LLM-guided Hierarchical Retrieval

## Abstract

Modern IR systems are increasingly tasked with answering complex, multi-faceted queries that require deep reasoning rather than simple keyword or semantic matching. While LLM based IR has shown great promise, the current retrieve-then-rerank paradigm inherits the limits of embedding-based retrieval, parametric generative approaches are difficult to adapt to new information, and long-in-context approaches that put the entire corpus in context are computationally infeasible for large document corpora due to the quadratic attention complexity. To this end, we introduce a hierarchical retrieval framework LATTICE that enables an LLM to reason and navigate a large corpus with logarithmic search complexity in the number of documents, achieved by imposing a semantic tree structure on the corpus. Our approach comprises two stages: (1) an offline process where we organize the document collection into a semantic hierarchy – we explore two LLM-driven strategies for this: a bottom-up agglomerative approach and a top-down divisive approach using multi-level summaries; (2) an online traversal stage where a "search LLM" navigates this tree. A central challenge in using LLMs for search is that the LLM's relevance judgments are *noisy, context-dependent, and unaware of the underlying hierarchy*, making it difficult to compare nodes across different branches and levels of the tree. To solve this, our traversal algorithm estimates calibrated latent relevance scores from the LLM's local outputs, which are combined into a path relevance metric to guide the search globally across the tree. Our training-free framework achieves state-of-the-art zero-shot performance on the reasoning-intensive BRIGHT (Su et al., 2024) benchmark (with up to 420K corpus size), demonstrating improvements of up to 9% in Recall@100 and 5% in nDCG@10. Moreover, compared to the highly specialized and fine-tuned SOTA method DIVER-v2 (Long et al., 2025), it achieves comparable results on BRIGHT subsets that use a static corpus for evaluation.

## 1 Introduction

The proliferation of Large Language Models (LLMs) has catalyzed a paradigm shift in Information Retrieval (IR), moving beyond simple fact-finding towards complex problem-solving that demands nuanced understanding and reasoning. Modern user queries often require not just keyword or semantic matching, but a deeper level of inference, categorized as reasoning-based retrieval (Su et al., 2024). For instance, a user might seek a solution to a coding bug by describing its behavior, or ask for the unit digit of a complex mathematical expression that requires applying a specific theorem. Answering such queries effectively means retrieving documents that help reason *through* the problem, a task for which traditional IR systems are poorly equipped.

Current LLM-based IR systems primarily fall into two paradigms, each with inherent drawbacks. The first, **Retrieve-then-Rerank**, employs a computationally cheap retriever (e.g., BM25 or dense retrieval) to fetch a broad set of candidate documents, which are then re-ordered by a more powerful but expensive LLM. Although scalable, this approach is constrained with the *limits of the initial retrieval stage* (Weller et al., 2025); if a crucial document is not captured in the initial candidate set, even a perfect reranker cannot recover it. Furthermore, the initial retrieval often relies on shallow semantic similarity, failing to perform the multi-step reasoning needed to identify relevant documents for complex queries.

The second paradigm, **Generative Retrieval (GenIR)**, uses the LLM itself to synthesize an answer. This can be **parametric**, where the corpus is stored implicitly in the model weights, making the

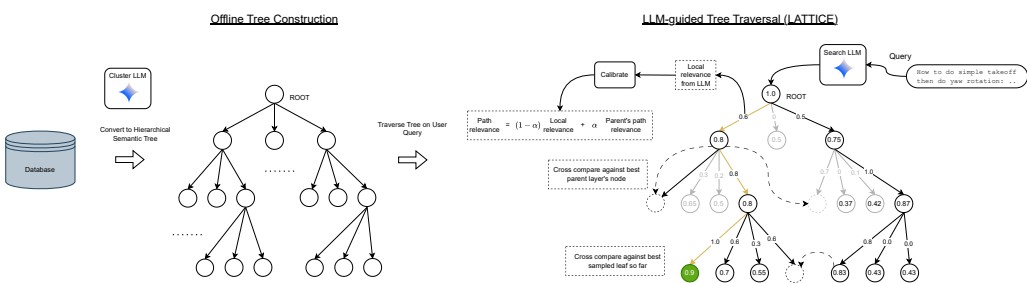

Figure 1: A high-level overview of our proposed framework, LATTICE. The process consists of two stages. (Left) In the offline stage, we organize an unstructured document corpus into a semantic tree. (Right) In the online stage, a search LLM performs a greedy, best-first traversal of this tree to find documents relevant to a user query. To guide the search, the algorithm computes a path relevance score at each step and uses a score calibration mechanism that compares nodes against high-relevance candidates from sibling branches and previously seen leaves, ensuring a globally coherent search. The search process is visualized for a real query in Figure 5.

system prone to hallucinations and difficult to update with new information. Alternatively, **long-context GenIR** places the entire corpus (or a large subset) explicitly into the LLM's context. While this allows the LLM to reason over the full text, it is computationally infeasible for a typical retrieval corpora, as the self-attention mechanism's quadratic/super-linear complexity leads to prohibitive costs and latency.

To overcome these limitations, we propose **LATTICE** (LArge language model guided Tree Traversal with Iterative Calibrated-score Estimation), a framework that combines the logarithmic search efficiency of hierarchical structures with the sophisticated reasoning capabilities of modern LLMs. Our method first organizes a document corpus into a semantic tree offline, with internal nodes represented by rich, LLM-generated textual summaries. Then, at query time, a search LLM navigates this semantic hierarchy using a greedy, best-first traversal, processing a beam of top candidates at each step. To ensure the search remains globally coherent, the traversal algorithm computes a path relevance score for each node by aggregating calibrated local scores from the LLM along the path from the root, allowing our method to robustly compare nodes across different branches and levels and efficiently reach the most relevant documents. Our main contributions are:

- We introduce a novel retrieval framework where an LLM directly performs the traversal of a semantic hierarchy, using its reasoning capabilities to guide the search path at each step, achieving state-of-the-art zero-shot results on the reasoning-intensive BRIGHT benchmark with improvements of up to 9% in Recall@100 and 5% in nDCG@10.
- We propose a robust traversal algorithm, that performs greedy search on a semantic tree using noisy LLM judgments.
- We design and compare two distinct, LLM-driven strategies for corpus organization: a bottom-up agglomerative clustering method and a top-down divisive summarization approach.

As LLMs become a fundamental unit of computation, the main goal of this paper is to demonstrate an LLM-native retrieval system that moves beyond the traditional application of LLMs in IR.

## 2 RELATED WORKS

### 2.1 LLMS FOR INFORMATION RETRIEVAL

**Retrieve-then-Rerank Paradigm.** The dominant paradigm in modern IR is a two-stage retrieve-then-rerank pipeline (Zhu et al., 2023). LLMs have excelled as powerful rerankers in this framework, applied in either pointwise (score each document independently) or listwise fashion (rank a list of documents) (Reddy et al., 2024; Sun et al., 2024). However, the overall performance is irreversibly bottlenecked by the quality of the initial retrieval stage (Rathee et al., 2025). In the retrieval stage, LLMs are increasingly used as backbones for dense embedding models (Luo et al., 2024; Lee et al., 2025), though this often involves adapting their autoregressive architecture for representation learning which is not directly aligned with their pre-training task.

**Generative Paradigms.** To overcome the limitations of the cascading pipeline, alternative paradigms have emerged. **Generative Retrieval**, such as the Differentiable Search Index (DSI) (Tay et al., 2022; Li et al., 2024), reframes IR as a sequence-to-sequence task, mapping a query directly to a document identifier. While conceptually elegant, these methods face challenges in scaling and updating the index (Pradeep et al., 2023). **Long-Context Retrieval** proposes placing the entire corpus into the LLM's context window (Lee et al., 2024a), but this remains computationally infeasible for even moderate-scale applications. Our work offers a middle ground, using a semantic hierarchy to structure the corpus, enabling an LLM to navigate it efficiently without the scalability / updatability issues of generative retrieval or the computational cost of long-context models.

## 2.2 HIERARCHICAL RETRIEVAL

**Vector Hierarchies.** Hierarchical structures have been long used to improve computationally efficiency in tasks with large output spaces, such as in hierarchical softmax for language modeling (Morin & Bengio, 2005) and in tree-based methods for extreme multi-label classification (Prabhu & Varma, 2014; Chang et al., 2020; Gupta et al., 2022). In vector search, algorithms like Hierarchical Navigable Small World (HNSW) (Malkov & Yashunin, 2018) use a multi-layered graph for efficient approximate nearest neighbor search, though this hierarchy is geometric rather than semantic.

**Textual Hierarchies.** More recently, models like RAPTOR (Sarthi et al., 2024) construct a semantic hierarchy by recursively clustering and summarizing text chunks from the bottom up. This creates a tree with nodes representing different levels of abstraction. However, RAPTOR relies on conventional embedding-based similarity search to traverse this tree. Our work differs fundamentally by employing an LLM as an *active traversal agent* during the online retrieval phase. Instead of a static vector comparison, our model uses in-context reasoning at each node to decide the optimal path, transforming retrieval into an intelligent navigation process.

## 2.3 AGENTIC AND REASONING-BASED IR

**Reasoning as a Pre-processing Step.** A common approach to incorporate reasoning in IR is through query expansion (QE) (Wang et al., 2023; Gao et al., 2023). In this setup, an LLM enriches the query with generated text or a chain-of-thought analysis before it is passed to a standard retrieval system. While effective, this treats reasoning as a discrete, pre-retrieval step, leaving the core search mechanism unchanged and often resulting in complex, multi-component pipelines (Long et al., 2025; Shao et al., 2025).

**Agentic Frameworks.** The emerging field of Agentic IR (Jin et al., 2025; Zhang et al., 2024) conceptualizes retrieval as a multi-step, goal-oriented process. However, current implementations typically involve an LLM agent calling an external, black-box search tool, making its success contingent on the tool's effectiveness. Similarly, Graph-RAG (Edge et al., 2024; Zhang et al., 2025) leverages LLMs to reason over pre-structured knowledge graphs, but the role of LLMs to retrieve information from these graphs are limited. Our work integrates the reasoning agent more deeply into the retrieval process itself. The LLM is not just a pre-processor or a tool-caller but the core search mechanism, more specifically, it is an agent whose environment is the corpus's semantic tree. The tree provides essential scaffolding, constraining the agent's action space to make the search tractable, while the agent's reasoning enables intelligent traversal decisions, offering a more fundamental fusion of reasoning and retrieval.

## 3 METHODOLOGY

We begin by formalizing the task setup and notations in Section 3.1, followed by a detailed description of the search procedure in Section 3.2, and ending with the tree construction procedures in Section 3.3.

## 3.1 SETUP

The fundamental task is **retrieval**: given a large corpus of $|D|$ documents, $D = \{d_1, d_2, \ldots, d_{|D|}\}$, and a complex natural language query $q$, the objective is to retrieve a ranked list of documents $D_{rel} \subseteq D$. We define the core components and notations of our framework as follows:

- **Semantic Tree:** The corpus is organized into a tree $T = (V, E)$, with a single root node, $v_{root}$.

- **Nodes** ($v \in V$)**:** The set of nodes $V$ is partitioned into leaf nodes $V_L$ (corresponding to documents) and internal nodes $V_I$ (representing conceptual groupings).

- **Edges** ($E$)**:** The set of directed edges $E \subset V \times V$ consists of ordered pairs $(u, v)$, where $u = \text{parent}(v)$. The set of immediate children of a node $u$ is denoted as $C(u)$.

- **Node Representation** ($\phi(v)$)**:** Every node $v \in V$ has a textual representation $\phi(v)$. For $v_l \in V_L$, $\phi(v_l)$ is its document's content. For $v_i \in V_I$, $\phi(v_i)$ is an LLM-generated summary of its children.

- **Search LLM** ($\mathcal{L}$)**:** For the purpose of this paper we assume that the search LLM can be abstracted out as a **listwise scoring function**. Given a query $q$ and a list of $k$ candidate nodes $[v_1, \ldots, v_k]$, it returns a list of real-valued scores (along with a reasoning trace):

$$\mathcal{L}(q, [\phi(v_1), \ldots, \phi(v_k)]) = [s_1, \ldots, s_k]$$

where $s_i \in [0, 100], i = 1, \ldots, k$, we further normalize $s_i$ such that it $\in [0, 1]$. A higher score implies higher preference. The prompt structure is detailed in Figure 7.

## 3.2 Tree Traversal

The core challenge in using an LLM for hierarchical search is that its relevance judgments are inherently noisy, context-dependent and unaware of the underlying hierarchy. The score assigned to a node depends on the query as well as on the other nodes present in the list of options provided to the LLM. On top of this, these scores are inherently noisy due to un-deterministic reasoning chain / inference of LLMs. This makes it difficult to compare the promise of a node in one branch against a node in a completely different branch or at a different level of the tree. Given a search query, the goal of our traversal algorithm is to prioritize the exploration of relevant nodes in the tree by predicting a **path relevance score**, $\hat{p}_{rel}(v)$, which converts these noisy, local signals into a globally coherent signal. The algorithm, depicted in Figure 1 and formalized in Algorithm 1, proceeds in following steps.

**1. Initialization.** The search begins with a max-priority queue, the **frontier** ($F$), which is initialized with the root node $v_{root}$. Its score is set to $\hat{p}_{rel}(v_{root}) \leftarrow 1.0$. We also initialize an empty **prediction set (Pred)** to store candidate leaf nodes and a history of all observed scores, ScoreHistory $\leftarrow \emptyset$.

**2. Beam Expansion.** In each of the $N$ iterations, we expand a beam of the top $B$ most promising nodes from the frontier $F$. These nodes are selected based on their current path relevance scores $\hat{p}_{rel}$.

**3. Slate Construction with Calibration.** For each node $v$ in the beam, we construct a slate for the search LLM to evaluate. This slate consists of the children of the current node, $C(v)$, and augment it with a set $Aug(v)$. The composition of $Aug(v)$ depends on the type of nodes being evaluated:

- If $C(v)$ are **internal nodes**, $Aug(v)$ consists of the top-scoring sibling of $v$ to provide a cross-reference across different branches.
- If $C(v)$ are **leaf nodes**, $Aug(v)$ consists of $\ell$ (a hyperparameter) leaf nodes sampled from Pred according to a probability distribution proportional to $e^{\hat{p}_{rel}(u)}$, anchoring the evaluation against the best candidates found so far and giving a chance for best scoring candidates to be evaluated again in a different context. We show in Figure 2 this is essential for the final ranking.

**4. Latent Score Estimation and Path Relevance Update.** After the search LLM $\mathcal{L}$ evaluates the slate and produces local scores, we perform a global calibration step before updating path relevance. We model the observed score $s_v^i$ for a node $v$ in a given slate $i$ as a linear transformation of an underlying, slate-independent **latent relevance score** $\hat{s}_v$:

$$s_v^i \approx a \cdot \hat{s}_v + b^i$$

where $a$ is a single global scale parameter and $b^i$ is a per-slate bias parameter. After each new slate is evaluated, we update our estimates for all latent scores $\{\hat{s}_v\}$, $a$, and biases $\{b^i\}$ by treating this as a

---

**Algorithm 1** LLM-guided Hierarchical Tree Traversal

---

1: **Parameters:** $q, T, \mathcal{L}, B, N, K, \alpha$
2: **Initialize:**
3: Frontier $F \leftarrow$ new MaxPriorityQueue(), Pred $\leftarrow \emptyset$
4: ScoreHistory $\leftarrow \emptyset$, LatentScores $\leftarrow \emptyset$
5: $\hat{p}_{rel}(v_{root}) \leftarrow 1.0$, $F$.push($v_{root}, \hat{p}_{rel}(v_{root})$)
6: **for** $i = 1$ **to** $N$ **do**
7:     Beam $\leftarrow$ Extract top $B$ nodes from $F$
8:     **for all** $v$ in Beam **do**
9:         Slate $\leftarrow C(v) + Aug(v)$
10:         LocalScores $[s_{v'}]_{v' \in \text{Slate}} \leftarrow \mathcal{L}(q, [\phi(v')]_{v' \in \text{Slate}})$
11:         Add $\{(\text{slate\_id}_i, v', s_{v'}) \mid v' \in \text{Slate}\}$ to ScoreHistory
12:     **end for**
13:     LatentScores $\leftarrow$ SolveMLE(ScoreHistory) {Minimize MSE to find all $\hat{s}_v$}
14:     **for all** $v$ in Beam that were just expanded **do**
15:         **for all** $v'$ in Slate **do**
16:             $\hat{s}_{v'} \leftarrow$ LatentScores$[v']$
17:             $\hat{p}_{rel}(v') \leftarrow \alpha \cdot \hat{p}_{rel}(\text{parent}(v')) + (1 - \alpha) \cdot \hat{s}_{v'}$
18:         **end for**
19:         **for all** $v'$ in $C(v)$ **do**
20:             **if** $v'$ is a leaf node **then**
21:                 Add $v'$ to Pred
22:             **else**
23:                 $F$.push($v', \hat{p}_{rel}(v')$)
24:             **end if**
25:         **end for**
26:     **end for**
27: **end for**
28: **return** Top-$K$ nodes from Pred sorted by $\hat{p}_{rel}$

---

Maximum Likelihood Estimation (MLE) problem. We find the parameters that minimize the Mean Squared Error (MSE) across all scores observed thus far:

$$\min_{a, \{\hat{s}_v\}, \{b^i\}} \sum_i \sum_{v \in \text{slate}^i} (s_v^i - (a \cdot \hat{s}_v + b^i))^2.$$

Note that without $a, b^i$ parameters $\hat{s}_v$ reduces to the mean of all the scores seen so far, we notice improved performance with this formulation as it can account for noise in scoring. Other objectives like margin-based losses or probabilistic models like Plackett-Luce could be applied, we found simple modified MSE optimization to be most consistent. The resulting latent score $\hat{s}_v$ is used to define the path relevance:

$$\hat{p}_{rel}(v) = \alpha \cdot \hat{p}_{rel}(\text{parent}(v)) + (1 - \alpha) \cdot \hat{s}_v$$

Here $\alpha$ is a hyperparameter in $[0, 1]$. After scoring, newly evaluated internal nodes are added to the frontier $F$, and leaf nodes are added to the prediction set Pred.

**5. Termination.** The algorithm terminates after $N$ iterations. The final output is the set of top-$K$ documents from Pred, ranked by their final path relevance scores.

## 3.3 OFFLINE SEMANTIC TREE CONSTRUCTION

The objective is to create a hierarchical structure $T = (V, E)$ where every leaf node $v \in V_L$ is connected to the root node $v_{root}$ via a single path and each node $v \in V$ is annotated with a textual $\phi(v)$. The maximum branching factor of any node is constrained by a hyperparameter, $M$ i.e. $|C(v)| \leq M \; \forall \; v \in V$. While our traversal algorithm can be adapted for more general Directed Acyclic Graph (DAG) structures, we focus on a strict tree for simplicity in this work. We now describe our bottom-up construction approach, which is conceptually similar to recursive clustering and summarization methods like RAPTOR (Sarthi et al., 2024).

### 3.3.1 APPROACH 1: BOTTOM-UP CLUSTERING AND SUMMARIZATION

This approach constructs the tree layer by layer, starting from the leaf nodes and iteratively clustering and summarizing them until a single root node is formed. To do this, we require two main components:

- An **embedding function**, $\mathcal{E} : \text{text} \to \mathbb{R}^d$, which maps a textual representation $\phi(v)$ to a $d$-dimensional vector. We use Gecko embeddings (Lee et al., 2024b) in our experiments.
- A **clustering function**, $\mathcal{C}$. Given a set of $n$ vectors $X = \{\mathbf{x}_1, \ldots, \mathbf{x}_n\}$, the function produces a partition $\{K_1, \ldots, K_m\}$ of $X$, such that for all $j \in \{1, \ldots, m\}$, $|K_j| \leq M$ and $K_i \cap K_j = \emptyset$ for $i \neq j$. This can be implemented via iterative application of standard clustering algorithms like spectral clustering.

The construction process, formalized in Algorithm 2, proceeds as follows:

**1. Initial Layer Formation.** The process begins with the set of leaf nodes, $V_L$. We form an initial set of parent nodes, $V_{\text{current}}$, one level above the leaves. This can be done in two ways:

- **From Scratch:** Apply the embedding and clustering functions to all documents to form the initial parent nodes.
- **Using Metadata:** For datasets where documents are passages from a smaller set of source articles (stackexchange sub-datasets in BRIGHT), we leverage this inherent structure. We form initial clusters by grouping all passages belonging to the same source document. If any of the resulting cluster contains more than $M$ passages, we further group nodes in the cluster based on location proximity in the source document until all sub-clusters satisfy the branching factor constraint. This metadata-driven approach often yields more semantically coherent initial groupings. Further implementation details are provided in Appendix B.3.

**2. Iterative Clustering and Summarization.** Starting with the initial set of parent nodes, $V_{\text{current}}$, we iteratively repeat a summarize-embed-cluster cycle. In each iteration, we first generate a textual summary $\phi(v)$ for each node in $V_{\text{current}}$, embed these new summaries, and cluster them to form the next, higher level of the tree.

**3. Termination.** We repeat this process until the number of nodes at the current level is less than or equal to $M$. These final nodes are assigned as the children of the root node, $v_{root}$, completing the tree.

### 3.3.2 APPROACH 2: TOP-DOWN DIVISIVE CLUSTERING

As an alternative to the agglomerative bottom-up method, we also explore a top-down divisive approach. Conceptually, this method is similar to hierarchical k-means, where we begin with a single cluster containing the entire document corpus and recursively partition it. The standard implementation would use an embedding and clustering function at each step. However, we observed that this can produce noisy, suboptimal clusters at the higher levels of the tree where partitions should be based on broad conceptual similarities rather than keyword overlap.

To address this, we employ an LLM as a more powerful clustering function. Since providing the entire corpus to an LLM is infeasible due to context limits, we introduce a prerequisite step: **hierarchical summarization**. For each leaf node $v_l$, we prompt an LLM to generate five summaries in increasing order of complexity (we quantify the complexity of a summary by its length, for e.g. first level of summary is 1-2 word, next is 3-4 words, and so on, more details in Section B.3.2), yielding a set of multi-level representations $\{\phi(v_l)^i\}_{i=1}^5$.

The top-down construction, detailed in Algorithm 4, proceeds as a recursive partitioning process:

**1. Initialization.** The process begins with a work queue containing the root node $v_{root}$, whose children are initially all leaf nodes $V_L$.

**2. Recursive Partitioning.** We iteratively process nodes from the queue. For each node $v$ to be partitioned, we first select an appropriate summary level $i$ for its leaf descendants (details in the

Section B.3.2). We then provide the set of unique summaries at that level to an LLM, prompting it to group them into $M$ conceptual topics.

**3. Node Creation and Re-assignment.** The LLM returns a description for each of the $M$ topics and a mapping from the unique input summaries to these topics. We create $M$ new internal nodes, assign them the topic descriptions, and partition the leaf descendants of $v$ among these new nodes according to the LLM's mapping. These $M$ new nodes become the children of $v$. Any new node that still contains more than $M$ leaves is added to the queue for further partitioning.

**4. Termination.** The process terminates when the queue is empty, meaning all internal nodes in the tree satisfy the maximum branching factor constraint.

## 4 EXPERIMENTS

### 4.1 EXPERIMENTAL SETUP

**Benchmark.** All experiments are conducted on the BRIGHT benchmark (Su et al., 2024), a collection of 12 reasoning-intensive retrieval tasks. The benchmark is specifically designed to evaluate deep reasoning and is composed of complex questions from diverse sources, including StackExchange, Leetcode, and TheoremQA, spanning topics from biology and economics to programming and mathematics.

**Evaluation Metrics.** We use two standard IR metrics to measure performance: **nDCG@10** (Normalized Discounted Cumulative Gain at 10) to evaluate the ranking quality of the top 10 results, and **Recall@100** to measure the comprehensiveness of the retrieval within the top 100 results.

**Baselines.** We compare LATTICE against several strong baselines.

- **SOTA Systems:** We compare against state-of-the-art systems like **DIVER-v1/v2** (Long et al., 2025), **RaDeR** (Das et al., 2025), **ReasonRank** (Liu et al., 2025) and **ReasonIR** (Shao et al., 2025), which are trained and highly specialized for the BRIGHT benchmark.

- **Controlled Reranking Baseline:** To ensure a fair, apples-to-apples comparison, we include a strong retrieve-then-rerank baseline **XRR2**[1](BM25 + Rerank) that uses the *same base LLM* (Gemini-2.5-flash) as our method. XRR2 first retrieves 100 candidates using BM25 with a GPT-4 expanded query and then reranks them using Gemini-2.5-flash model for total 5 iterations. This allows us to isolate the performance gains attributable to directly using an LLM to search the space versus just reranking a small retrieved corpus.

**Implementation Details.** For all LLM-driven components of our method (tree construction, summarization, and online search), we use **Gemini-2.5-flash** (Comanici et al., 2025). For the online traversal, we set the path relevance momentum to $\alpha = 0.5$, the number of iterations to $N = 20$, $\ell = 10$ and the beam size to $B = 2$. This configuration results in approximately 250 documents being evaluated by the LLM per query. For tree construction, the maximum branching factor was set to $M \sim 10 - 20$. For datasets derived from StackExchange, we employed the bottom-up clustering method; for all others, we used the top-down divisive approach. Our method, LATTICE, is evaluated in a strictly zero-shot setting, without any fine-tuning or ensembling with any other method for the BRIGHT benchmark tasks. Further details are provided in Appendix B.

### 4.2 PERFORMANCE ON THE BRIGHT BENCHMARK

**Ranking Performance (nDCG@10)** We present main ranking results on the BRIGHT benchmark in Table 1. On the seven StackExchange datasets, which use a standard static corpus, LATTICE achieves an average nDCG@10 of **51.6**, significantly outperforming the controlled reranking baseline's score of **47.4**. Furthermore, our zero-shot performance is highly competitive with the fine-tuned SOTA, Diver-v2 (**52.2**), and even achieves the best results in several sub-domains like Economics and Robotics. On the 3/5 Coding and Theorem-based tasks (LeetCode, AoPS & TheoremQ),

---

[1]https://github.com/jataware/XRR2/tree/main

| Method | Fine-tuned | StackExchange | | | | | | | | Coding | | | Theorem-based | | | | Avg. |
|---|---|---|---|---|---|---|---|---|---|---|---|---|---|---|---|---|---|
| | | Avg. | Bio. | Earth. | Econ. | Psy. | Rob. | Stack. | Sus. | Avg. | Leet.* | Pony | Avg. | AoPS* | ThQ.* | ThT. | |
| | | | | | | Retriever with GPT-4 REASON-query | | | | | | | | | | | |
| BM25 | ✗ | 34.8 | 53.6 | 54.1 | 24.3 | 38.7 | 18.9 | 27.7 | 26.3 | 18.4 | 19.3 | 17.6 | 14.6 | 3.9 | 19.2 | 20.8 | 27.0 |
| SBERT | ✗ | 18.2 | 18.5 | 26.3 | 17.5 | 27.2 | 8.8 | 11.8 | 17.5 | 17.3 | 24.3 | 10.3 | 16.9 | 5.0 | 22.3 | 23.5 | 17.7 |
| gte-Qwen1.5-7B | ✗ | 28.4 | 35.5 | 43.1 | 24.3 | 34.3 | 15.4 | 22.9 | 23.9 | 15.3 | 25.4 | 5.2 | 22.6 | 4.6 | 28.7 | 34.6 | 24.8 |
| OpenAI | ✗ | 27.7 | 35.2 | 40.1 | 25.1 | 38.0 | 13.6 | 18.2 | 24.2 | 15.5 | 24.5 | 6.5 | 18.1 | 7.7 | 22.9 | 23.8 | 23.3 |
| Google | ✗ | 30.2 | 36.4 | 45.6 | 25.6 | 38.2 | 18.7 | 29.5 | 17.9 | 17.4 | 31.1 | 3.7 | 22.7 | 10.0 | 27.8 | 30.4 | 26.2 |
| ReasonIR-8B | ✓ | 33.1 | 43.6 | 42.9 | 32.7 | 38.8 | 20.9 | 25.8 | 27.5 | 25.5 | 31.5 | 19.6 | 25.4 | 7.4 | 33.1 | 35.7 | 29.9 |
| RaDeR-7B | ✓ | 30.1 | 36.1 | 42.9 | 25.2 | 37.9 | 16.6 | 27.4 | 25.0 | 23.3 | 34.8 | 11.9 | 31.0 | 12.0 | 37.7 | 43.4 | 29.2 |
| DIVER | ✓ | 35.8 | 51.9 | 53.5 | 29.5 | 41.2 | 21.4 | 27.5 | 26.1 | 22.6 | 33.5 | 11.7 | 29.5 | 9.5 | 39.3 | 39.7 | 32.0 |
| | | | | | | Retrieve-then-rerank | | | | | | | | | | | |
| ReasonIR | ✓ | 41.7 | 59.8 | 53.2 | 32.0 | 43.6 | 28.8 | 38.7 | 36.0 | **34.0** | 33.2 | 34.8 | 29.4 | 7.9 | 32.6 | 47.7 | 37.3 |
| DIVER v1 | ✓ | 46.1 | 62.2 | 58.7 | 34.4 | 52.9 | 35.6 | 36.5 | 42.9 | 32.1 | **38.9** | 25.4 | 37.1 | 18.3 | 40.0 | 53.1 | 41.5 |
| ReasonRank | ✓ | 46.8 | 62.7 | 55.5 | 36.7 | 54.6 | 35.7 | 38.0 | 44.8 | 27.5 | 29.5 | 25.6 | 35.5 | 14.4 | 42.0 | 50.1 | 40.8 |
| XRR2 | ✗ | 47.4 | 63.1 | 58.2 | 38.5 | 52.9 | 37.1 | 37.6 | 44.6 | 28.4 | 21.9 | **35.0** | 31.8 | 15.7 | 34.4 | 45.5 | 40.3 |
| DIVER v2 | ✓ | **52.2** | **68.0** | **62.5** | 42.0 | **58.2** | 41.5 | **44.3** | **49.2** | 33.8 | 34.8 | 32.9 | **38.6** | **19.1** | 44.3 | 52.6 | **45.7** |
| | | | | | | LLM-guided Hierarchical Retrieval | | | | | | | | | | | |
| LATTICE | ✗ | 51.6 | 64.4 | 62.4 | **45.4** | 57.4 | **47.6** | 37.6 | 46.4 | 26.9 | 19.9 | 34.0 | 30.0 | 12.0 | 30.1 | 47.8 | 42.1 |

Table 1: nDCG@10 performance of various retrievers and rankers on the BRIGHT benchmark. **Bold** represents overall best numbers, underline represents best numbers among zero-shot methods, * denotes subsets with dynamic corpus.

our method's performance is noticably lower than the baselines. This is attributable to a specific benchmark artifact: the use of a query-dependent dynamic corpus, where a unique large list (can be > 10K) of documents (which are potential positives) is excluded from the search space. While we prune the excluded leaf nodes at query time, the pre-computed summaries ($\phi(v)$) of their parent nodes do not update dynamically. Consequently, these summaries often misguide the traversal (please see Figure 6, Section C.2). In contrast, retrieve-then-rerank pipelines can simply filter excluded documents from their candidate list post-retrieval without penalty. We would like to note that most real-world IR systems operate on a query-independent corpus.

**Retrieval Comprehensiveness (Recall@100)** As illustrated in Figure 3, our method demonstrates superior overall retrieval comprehensiveness. On average, LATTICE achieves a Recall@100 of **74.8**, outperforming both the BM25 baseline (65.3) and the specialized ReasonIR-8B model (70.8). This strong performance is consistent across the majority of subsets, with our method achieving the highest recall in four of the seven domains, including Economics and Psychology.

**Cost-Performance Analysis.** To analyze the computational cost of our method, we compare the trade-off between performance (nDCG@10) and cost (measure in number of tokens processed by the LLM) against two retreive-then-rerank baselines using the gemini-2.5-flash as the ranking LLM and varying top-k predictions from the retriever. Figure 3 plots this relationship for the Robotics subset. While the reranking baselines exhibit diminishing returns, LATTICE's performance scales far more effectively on this subset. The performance initially remains flat as the model needs to take atleast tree height number of slate comparisons to reach a leaf node. This shows promise that our guided hierarchical search can be more efficient use of the LLM's computational budget than reranking a long, flat list of documents, where many of the tokens are spent on irrelevant candidates.

## 5 ANALYSIS

**Effect of # Cross-Branch Calibration ($\ell$).** Figure 2 shows the impact of including $\ell$ top-scoring nodes from sibling branches in the leaf slates on bio subset. The results demonstrate that this calibration is critical for effective search. The baseline with no calibration ($\ell = 0$) performs significantly worse and fails to improve with more search iterations. Performance consistently increases with $\ell$, with substantial gains from $\ell = 1$ to $\ell = 5$. The gains diminish after $\ell = 5$.

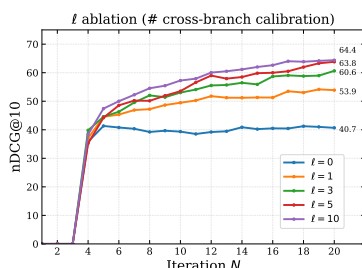

Figure 2: nDCG@10 vs. $\ell$.

**Impact of Method Components** To quantify the contribution of each component of LATTICE, we conduct a detailed ablation study with results presented in Table 2. We compare our full method against several variants: a version without score calibration (using raw LLM scores), one without path relevance (disabling path smoothing with $\alpha = 0$), one with zero reasoning budget to the LLM,

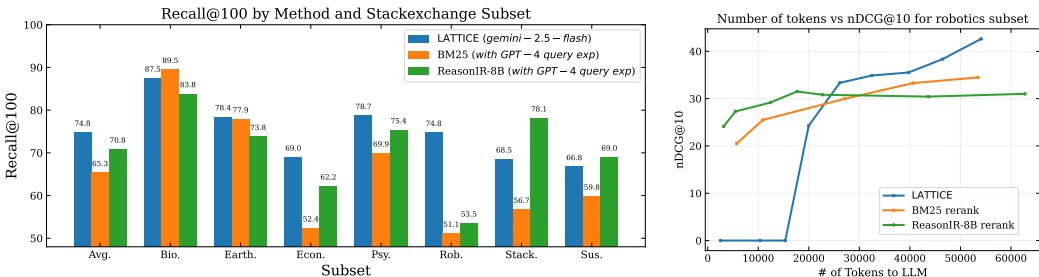

Figure 3: Results on retrieval (Recall@100) comprehensiveness (***left***) and nDCG@10 vs. token usage on Robotics dataset (***right***).

| Configuration | Avg. | Bio. | Earth. | Econ. | Psy. | Rob. | Stack. | Sus. |
|---|---|---|---|---|---|---|---|---|
| **LATTICE (Full Method)** | 51.57 | 64.38 | 62.36 | 45.37 | 57.35 | 47.57 | 37.58 | 46.35 |
| − No Score Calibration ($\hat{s}_v = $ last $s_v^i$) | 49.36 | 64.45 | 58.98 | 44.27 | 54.41 | 46.70 | 32.93 | 43.80 |
| − No Path Relevance ($\alpha = 0$) | 48.62 | 63.62 | 55.89 | 41.90 | 52.99 | 42.14 | 40.68 | 43.09 |
| − No Reasoning (thinking_budget= 0) | 49.33 | 63.69 | 57.32 | 43.77 | 57.33 | 45.73 | 33.16 | 43.95 |
| + Rerank Top-100 predictions | 48.54 | 62.42 | 59.10 | 42.05 | 54.33 | 45.26 | 34.33 | 42.29 |

Table 2: Ablation study on the core components of our traversal algorithm, evaluated across all StackExchange subsets of the BRIGHT benchmark. All values are nDCG@10.

and a final version where we add a reranking stage to our top 100 output. Interestingly, adding a final reranking step is detrimental, we hypothesize that our method's approach of decomposing a single complex ranking task into a sequence of smaller, high-fidelity local decisions produces a more accurate global ranking than a single, high-complexity reranking step over a large candidate list. Disabling path relevance smoothing causes the next largest degradation, followed by removing either the LLM's reasoning or score calibration mechanism also reduces the average score by over 2.2 nDCG points.

**Beam Size vs. Search Iterations.** Figure 4 presents a budget-matched analysis of beam size ($B$) versus search iterations ($N$), where the total number of node expansions ($B \times N$) is kept roughly constant. The results clearly indicate that for a fixed computational budget, prioritizing search depth (more iterations) over breadth (a larger beam) is the superior strategy. The configurations with smaller beams, $B = 1$ and $B = 2$, achieve the highest final nDCG@10 scores but are more sequential. This validates our choice of using a small beam size ($B = 2$) with a moderate number of iterations.

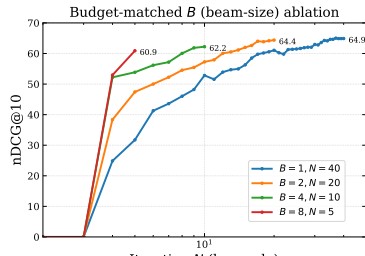

Figure 4: nDCG@10 vs. beam-size.

**Impact of Tree Construction Strategy** We investigate the impact of the tree construction strategy on two representative datasets in Table 3. The results show that aligning the tree construction method with the corpus's underlying structure is critical for zero-shot performance. For the Biology dataset, which is composed of passages from larger source documents, the bottom-up approach is superior, improving nDCG@10 by over 9 points. We hypothesize that this is because it leverages the inherent

| Biology | | TheoT. | |
|---|---|---|---|
| nDCG@10 | R@100 | nDCG@10 | R@100 |
| Bottom-Up Tree | | | |
| **64.38** | **87.53** | 35.89 | 61.82 |
| Top-Down Tree | | | |
| 55.22 | 67.31 | **47.85** | **73.91** |

Table 3: Tree construction comparison.

part-whole relationships in the data. Conversely, for the TheoT. dataset, which is a collection of distinct documents under a high-level topic, the top-down approach excels, improving nDCG@10 by nearly 12 points. We hypothesize that this method is better suited to discovering the latent conceptual clusters among independent documents.

## REPRODUCIBILITY STATEMENT

We are committed to ensuring the reproducibility of our work. The core algorithms for our traversal and tree construction methods are formally described in Section 3, with detailed pseudocode provided in Algorithms 1, 2, and 4. All hyperparameters, benchmark details, evaluation metrics, and baselines are specified in our experimental setup (Section 3.1, B). We have included comprehensive implementation details and ablation studies in Section 4 and the Appendix to allow our results to be fully reproduced. Upon acceptance of this paper, we will publicly release our source code, the constructed semantic trees used in our experiments, and our evaluation logs.

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

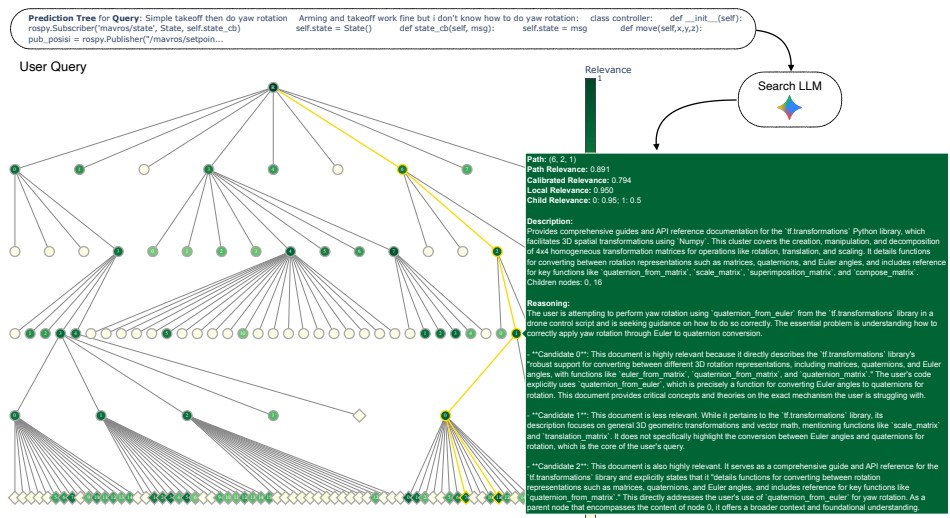

Figure 5: An illustration of the search process of LATTICE for a real query from the BRIGHT benchmark. The color of each node corresponds to its computed path relevance, highlighted yellow path shows the path to ground-truth documents. The search LLM makes a step-by-step decision at each internal node to determine which branch to explore next. The expanded callout provides a "glass box" view into one such decision, detailing the LLM's explicit reasoning process as it scores the children nodes.

# A  LIMITATIONS AND FUTURE WORK

Our work introduces a novel framework for hierarchical retrieval, but it also presents several avenues for future research. One of the limitation of our current approach is the use of a **static semantic tree**. As demonstrated in our experiments on dynamic corpora, the pre-computed summaries of internal nodes do not update when leaf nodes are filtered, which can occasionally misguide the search. Future work could explore methods for efficient, localized updates to the tree's summaries, allowing the hierarchy to adapt to a changing corpus without the need for a full reconstruction.

Second, the **offline tree construction process**, while a one-time cost, can be computationally intensive for extremely large corpora due to the repeated use of LLMs for clustering and summarization. Research into more efficient construction methods, perhaps by combining traditional clustering for the lower levels with LLM-based summarization for only the top, most abstract layers, could further improve scalability.

Finally, our traversal algorithm opens up new research directions. The score calibration method, while effective, uses a simple linear model. More sophisticated probabilistic models, could be explored for even more robust **latent score estimation**. Furthermore, while our greedy, best-first traversal is effective in a zero-shot setting, the entire process could be framed as a reinforcement learning problem, where the search LLM is an agent trained to optimize a policy for navigating the tree to maximize retrieval rewards. We believe that exploring these directions will further establish hierarchical, LLM-driven navigation as a powerful new paradigm in information retrieval.

# B  IMPLEMENTATION DETAILS

## B.1  HYPERPARAMETERS

This section provides a detailed list of all hyperparameters and implementation choices used in our experiments to ensure full reproducibility.

### B.1.1  OFFLINE TREE CONSTRUCTION

- **Maximum Branching Factor ($M$):** We set the maximum number of children for any node to $M = 10 - 20$.

- **Embedding Model ($\mathcal{E}$):** We use `gecko` (Lee et al., 2024b) embeddings to generate vector representations for the clustering steps.

- **Clustering Algorithm ($\mathcal{C}$):** Our implementation uses an iterative spectral clustering (Ng et al., 2001) algorithm to partition nodes into at most $M$ clusters at each level of the hierarchy.

- **Summarization LLM:** We use `Gemini-2.5-flash` for all summarization tasks (both for internal nodes in the bottom-up method and for the multi-level document summaries in the top-down method). The exact prompt template used is detailed in Appendix D.

- **Top-Down Summary Levels:** For the top-down method, we generate 5 levels of hierarchical summaries for each document.

### B.1.2 ONLINE TRAVERSAL

- **Search LLM ($\mathcal{L}$):** We use `Gemini-2.5-flash` as the search agent that performs the listwise scoring. The prompt structure is provided in Appendix D.

- **Number of Iterations ($N$):** We run the search for $N = 20$ iterations for all main experiments.

- **Beam Size ($B$):** We use a beam size of $B = 2$ for parallel node expansion in each iteration.

- **Path Relevance Momentum ($\alpha$):** The smoothing factor for the path relevance score is set to $\alpha = 0.5$.

- **Calibration Nodes ($l$):** We augment each leaf slate with $\ell = 10$ cross-branch leaf nodes for calibration, based on our ablation study.

- **Reasoning Budget:** The default "thinking budget" for the LLM's reasoning step is set to `-1`, meaning the model gets to decide how long it wants to thin.

- **MLE Solver:** The latent scores are updated after each batch of slate evaluations. The MSE loss is minimized using the Adam optimizer with a learning rate of $10^{-2}$ for `100` steps.

**Usage of LLMs**   During the preparation of this manuscript, LLM were used as a collaborative writing assistant to aid with drafting, refining prose for clarity and conciseness, and structuring arguments; all core ideas, experiments, and analyses were conducted by the authors.

### B.2 DATASET DETAILS

All experiments are conducted on the BRIGHT benchmark (Su et al., 2024), a comprehensive collection of 12 datasets designed to evaluate reasoning-intensive retrieval. A summary of the statistics for each subset is provided in Table 4.

The datasets exhibit two key characteristics relevant to our work. First, the StackExchange subsets are composed of passages derived from longer source documents. We leverage this structure for our metadata-based initial clustering in the bottom-up tree construction method. Second, the Coding and Theorem-based datasets (excluding Pony and TheoremQA Theorems) utilize a **query-dependent corpus**, where a unique list of documents (often >10k) must be excluded from the search space for each query. This feature, discussed in our main results analysis, poses a unique challenge for static index structures like our semantic tree.

### B.3 TREE CONSTRUCTION

### B.3.1 BOTTOM-UP

The Bottom-up tree constructions algorithms are defined in Alogirthm 2, 3.

### B.3.2 TOP-DOWN

The Top-down tree constructions algorithm is defined in Algorithm 4, the two subroutines used are described below.

| Dataset Subset | # Queries | Corpus Size ($\mathcal{D}$) | Avg. Doc Length |
|---|---|---|---|
| *StackExchange* | | | |
| Biology | 103 | 57,359 | 83.6 |
| Earth Science | 116 | 121,249 | 132.6 |
| Economics | 103 | 50,220 | 120.2 |
| Psychology | 101 | 52,835 | 118.2 |
| Robotics | 101 | 61,961 | 121.0 |
| Stack Overflow | 117 | 107,081 | 704.7 |
| Sustainable Living | 108 | 60,792 | 107.9 |
| *Coding* | | | |
| LeetCode | 142 | 413,932 | 482.6 |
| Pony | 112 | 7,894 | 98.3 |
| *Math* | | | |
| AoPS | 111 | 188,002 | 250.5 |
| TheoremQA-Q | 194 | 188,002 | 250.5 |
| TheoremQA-T | 76 | 23,839 | 354.8 |

Table 4: Statistics for the 12 subsets of the BRIGHT benchmark used in our experiments.

---

**Algorithm 2** Bottom-Up Tree Construction

---

1: **Parameters:** Corpus $D$, $\mathcal{E}$, $\mathcal{C}$, Summarize LLM, $M$, Optional InitialClusters
2: **Initialize:** $V_L \leftarrow \{\text{Node}(d) \mid d \in D\}$, $V \leftarrow V_L$, $E \leftarrow \emptyset$
3: **if** InitialClusters is provided **then**
4:    $V_{\text{current}} \leftarrow \text{CreateNodesFromClusters}(V_L, \text{InitialClusters}, V, E)$
5: **else**
6:    Embeddings $\leftarrow \{\mathcal{E}(\phi(v)) : v \in V_L\}$
7:    Clusters $\leftarrow \mathcal{C}(\text{Embeddings})$
8:    $V_{\text{current}} \leftarrow \text{CreateNodesFromClusters}(V_L, \text{Clusters}, V, E)$
9: **end if**
10: **while** $|V_{\text{current}}| > M$ **do**
11:    {Summarize the current layer before clustering}
12:    **for all** $v$ in $V_{\text{current}}$ **do**
13:       $\phi(v) \leftarrow \text{Summarize}(\{\phi(c) \mid c \in C(v)\})$
14:    **end for**
15:    $V_{\text{next\_layer}} \leftarrow \emptyset$
16:    Embeddings $\leftarrow \{\mathcal{E}(\phi(v)) : v \in V_{\text{current}}\}$
17:    Clusters $\leftarrow \mathcal{C}(\text{Embeddings})$
18:    $V_{\text{next\_layer}} \leftarrow \text{CreateNodesFromClusters}(V_{\text{current}}, \text{Clusters}, V, E)$
19:    $V_{\text{current}} \leftarrow V_{\text{next\_layer}}$
20: **end while**
21: $v_{root} \leftarrow \text{NewInternalNode}()$, $\phi(v_{root}) \leftarrow ""$
22: $C(v_{root}) \leftarrow V_{\text{current}}$
23: $V \leftarrow V \cup \{v_{root}\}$, $E \leftarrow E \cup \{(v_{root}, c) \mid c \in C(v_{root})\}$
24: **return** Tree $T = (V, E)$

---

The **SelectSummaryLevel** function implements a heuristic to find the optimal summary granularity for a given set of leaf nodes. It begins with the most abstract summary level ($i = 1$) and iteratively checks the number of unique summaries, selecting the first level $i$ where the count of unique summaries is sufficient for meaningful clustering (e.g., greater than $M$) while remaining under a maximum token limit for the LLM context.

The **ClusterLLM** function is realized via a structured prompt (see 9. The LLM is provided with the list of unique summaries and tasked with grouping them into $M$ coherent conceptual clusters. The prompt instructs the model to first generate a short, descriptive title for each of the $M$ clusters, and

---

**Algorithm 3** CreateNodesFromClusters Subroutine

---

1: **function** CreateNodesFromClusters($V_{\text{source}}$, Clusters, $V$, $E$)
2: **Input:**
3:    $V_{\text{source}}$: The set of nodes in the layer to be clustered.
4:    Clusters: The partition of $V_{\text{source}}$'s embeddings from $\mathcal{C}$.
5:    $V, E$: The global node and edge sets for the tree (passed by reference).
6: **Initialize:** $V_{\text{new\_layer}} \leftarrow \emptyset$
7: **for all** cluster $K$ in Clusters **do**
8:    $v_{new} \leftarrow$ NewInternalNode()
9:    $C(v_{new}) \leftarrow \{v \in V_{\text{source}} \mid v \in K\}$
10:    $V \leftarrow V \cup \{v_{new}\}$
11:    $E \leftarrow E \cup \{(v_{new}, c) \mid c \in C(v_{new})\}$
12:    $V_{\text{new\_layer}} \leftarrow V_{\text{new\_layer}} \cup \{v_{new}\}$
13: **end for**
14: **return** $V_{\text{new\_layer}}$

---

**Algorithm 4** Top-Down Divisive Tree Construction

---

1: **Parameters:** Corpus $D$, Summarize LLM, Cluster LLM, Max branching factor $M$
2: **Initialize:**
3: For each document $d_l \in D$, generate multi-level summaries $\{\phi(v_l)^i\}_{i=1}^{5}$.
4: $V_L \leftarrow \{\text{Node}(d) \mid d \in D\}$, $V \leftarrow V_L$
5: $v_{root} \leftarrow$ NewInternalNode(), $C(v_{root}) \leftarrow V_L$
6: $V \leftarrow V \cup \{v_{root}\}$, $E \leftarrow \{(v_{root}, c) \mid c \in V_L\}$
7: PartitionQueue $\leftarrow$ new Queue()
8: **if** $|V_L| > M$ **then**
9:    PartitionQueue.enqueue($v_{root}$)
10: **end if**
11: **while** PartitionQueue is not empty **do**
12:    $v \leftarrow$ PartitionQueue.dequeue()
13:    LeafDescendants $\leftarrow$ GetLeafDescendants($v, T$)
14:    $i \leftarrow$ SelectSummaryLevel(LeafDescendants)
15:    UniqueSummaries $\leftarrow$ unique($\{\phi(c)^i \mid c \in \text{LeafDescendants}\}$)
16:    TopicDescs, Mapping $\leftarrow$ ClusterLLM(UniqueSummaries, $M$)
17:    NewChildren $\leftarrow \emptyset$
18:    **for** $j = 1$ **to** $M$ **do**
19:      $v'_j \leftarrow$ NewInternalNode(), $\phi(v'_j) \leftarrow$ TopicDescs[$j$]
20:      $V \leftarrow V \cup \{v'_j\}$, NewChildren $\leftarrow$ NewChildren $\cup \{v'_j\}$
21:    **end for**
22:    ReassignChildren(LeafDescendants, Mapping, NewChildren, T)
23:    $E \leftarrow E \setminus \{(v, c) \mid c \in C(v)\}$ {Disconnect old children}
24:    $C(v) \leftarrow$ NewChildren
25:    $E \leftarrow E \cup \{(v, c) \mid c \in \text{NewChildren}\}$ {Connect new children}
26:    **for all** $v'_j$ in NewChildren **do**
27:      **if** $|C(v'_j)| > M$ **then**
28:        PartitionQueue.enqueue($v'_j$)
29:      **end if**
30:    **end for**
31: **end while**
32: **return** Tree $T = (V, E)$

---

then to output a mapping from each input summary to one of these cluster titles. The final output is a structured object containing the $M$ topic descriptions (which become the $\phi(v)$ for the new nodes) and the mapping.

## C SUBJECTIVE ANALYSIS

### C.1 SAMPLE SCORING RESPONSE FROM LLM

To provide a more intuitive understanding of our method, Figure 5 presents a qualitative case study of the search process for a real query from the BRIGHT benchmark. The user query is a code snippet asking about "yaw rotation," a complex 3D graphics problem. The figure visualizes the semantic tree and the traversal path taken by LATTICE (highlighted in yellow) to successfully locate a relevant document deep within the hierarchy.

The expanded callout provides a "glass box" view into the search LLM's reasoning at a critical decision point. The LLM's generated **Reasoning** explicitly connects the user's query to the node's topic, noting that the user is "attempting to perform yaw rotation using quaternion_from_euler." It then performs a detailed, comparative evaluation of the children nodes. It correctly identifies Candidate 1 as highly relevant because it discusses "support for converting between different 3D rotation representations, including matrices, quaternions, and Euler angles," which directly addresses the user's problem. This example demonstrates that our method does not rely on shallow semantic similarity; instead, the search is an active process guided by the LLM's deep, step-by-step reasoning about the query in the context of the corpus hierarchy.

### C.2 SEARCH FAILURE ON DYNAMIC CORPUS

Figure 6 provides a qualitative case study of a search failure, visually demonstrating the primary challenge our method faces on datasets with a dynamic corpus. The figure shows the search tree for a random query from the AoPS dataset. Red edges indicate leaf nodes that were dynamically excluded for this specific query, while the yellow path highlights the ideal traversal route to the ground-truth document.

As the figure shows, the search agent correctly follows the ground-truth path for the first two levels. However, it then reaches an internal node whose pre-computed summary is now misleading; the summary was generated based on all of its children, including the large number that have since been pruned from the search space (the red nodes). This inaccurate, stale summary causes the search LLM to make an incorrect judgment, deviating from the correct path and ultimately failing to retrieve the relevant document. This example visually confirms the specific failure mode of a static hierarchical index when faced with a dynamic corpus, reinforcing the quantitative analysis in our main results section.

## D PROMPTS

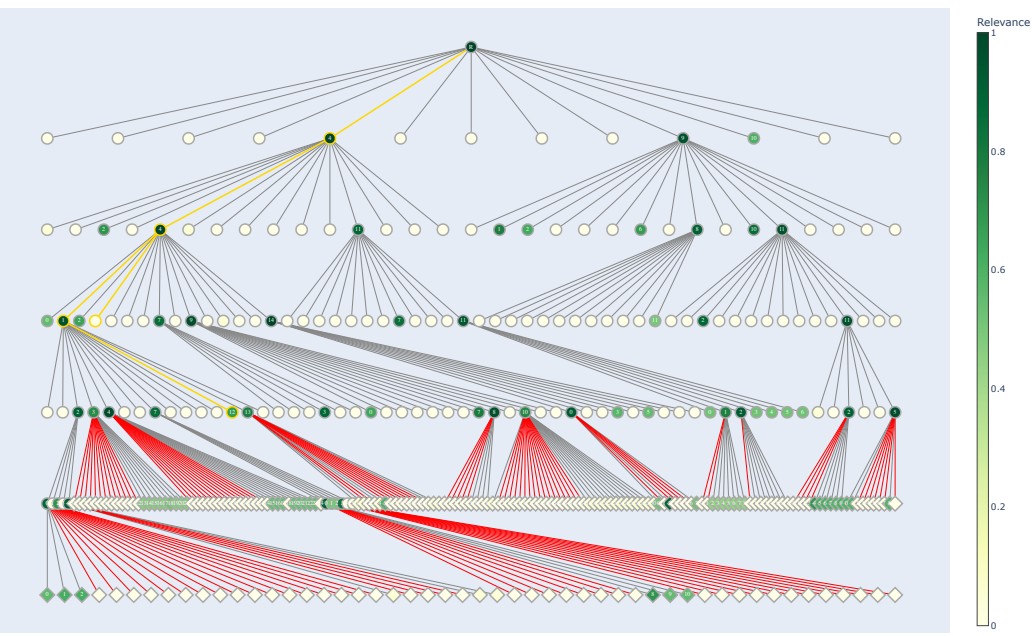

Figure 6: Search failing due to dynamically excluded search corpus, red edges denote excluded leaf nodes, gold edges denote ground-truth path

```
You are an intelligent search agent navigating a hierarchical semantic tree of topics. Your mission is to
predict the most promising candidates to find the answer to the user's query using the relevance definition
 below.

**Relevance Definition:** {relevance_defintion}

---

## USER QUERY

{query}

---

## CANDIDATES

Here are the candidates, each is identified by a unique `node_id` provided at the very start in [] (e.g.,
[0]).

{child_node_options}

---

## YOUR EVALUATION TASK
1.  First, identify the essential problem in the query.
2.  Think step by step to reason about why each candidate is relevant or irrelevant (based on the relevance
 definition). Provide this analysis in the `reasoning` field.
3.  Rank these passages based on their relevance to the query. Provide your ranking in the `ranking` field.
4.  Assign a relevance score from 0 to 100 (based on the relevance definition and the ranking). Provide
relevances in the `relevance_scores` field.

---

## OUTPUT FORMAT
You must provide your response as a single, clean JSON object. The JSON should have three keys: `reasoning
`, `ranking`, and `relevance_scores`.

* `reasoning`: This must be a **string**.
* `ranking`: This must be an **array of integers** representing the order of the candidates.
* `relevance_scores`: This must be an **array of arrays** where each inner array contains [node_id,
relevance_score]. For example: [[0, 85], [1, 92], [2, 73]].

---

## YOUR RESPONSE
```

Figure 7: Prompt template used in our experiments for scoring a list of nodes for $\mathcal{L}$.

```
You are an expert in information retrieval and keyword generation. Your task is to analyze a provided list
of informational passages and generate a hierarchically sorted list of search keywords for each passage,
strictly adhering to the 5-level rubric below.

## Keyword Generation Rules (5 Levels):

Level 1: 1-2 Word, Core Subject / Domain (Broadest)
Meaning: The absolute fundamental, overarching subject area or discipline.
Characteristics: Only 1 to 2 word, very high-level (e.g., "Technology", "Science", "History")

Level 2: 3-4 Word, General Topic / Sub-domain
Meaning: Narrows Level 1; the specific major topic or branch within the broader field.
Characteristics: Only 3 to 4 words, still general but more focused

Level 3: 4-6 Word, Key Concepts / Main Themes
Meaning: The central ideas, significant concepts, or primary themes directly discussed.
Characteristics: Only 4 to 6 words, core messages, primary subjects, often main sections

Level 4: 7-10 Word, Very Concise Passage Summary
Meaning: A very short, concise summary of what the entire passage is about. This should encapsulate the
essential idea or purpose of the passage.
Characteristics: Only 7 to 10 words

Level 5: 11-20 Word, Concise Passage Summary (Most Specific)
Meaning: A concise summary but more descriptive than level 4 of what the entire passage is about. This
should encapsulate the main idea or purpose of the passage.
Characteristics: A single sentence, 11 to 20 words.

### General Keyword Requirements:

- All keywords must be actionable terms or phrases a user would realistically search.
- Ensure comprehensive coverage of the passage's content across all 5 levels.

## Output Format

Your output must be a single JSON object. This object will contain a top-level key: "passages_keywords".
The value associated with this key must be a JSON array. Each element in this array will be an object with
two keys:
"passage_id": An integer that exactly matches the "id" from the corresponding input passage.
"hierarchical_keywords": A JSON array of strings of length 5. Each string represents a hierarchical level (
Level 1 at index 0, Level 2 at index 1, and so on).

## List of Input Passages:

{desc_list}
```

Figure 8: Prompt template used in our experiments for generating multi-level keywords to be used in top-down tree construction.

```
You are an expert data analyst and taxonomist. Your task is to analyze a list of keywords and their
associated counts which indicate how many that keyword appears in the corpus.

## Goal
- Group the following keywords into **k** semantically coherent and **well-balanced** (i.e. each cluster
should aim to contain similar weighted count) clusters, where k is between [{min_k}, {max_k}]. The primary
basis for grouping must be the **topic and meaning** of the keywords.
- Use the provided count as a measure of each keyword's **importance or popularity**. This weight should
help you decide which topics are most significant.
- Try to always maximize the number of clusters but **without** sacrificing the quality of the clustering,
**quality of clustering is paramount**.

For every cluster, generate:
* A descriptive `cluster_name`.
* An information-dense `cluster_description` summarizing the core themes.
* A list of all input `keywords` that constitute this cluster or apply to this cluster.

## Input Data
Here is the list of keywords and their importance counts:

{keywords_list_with_count}

## Desired Output Format
Your final output must be a single JSON object, with no other text or explanation. The JSON object must
have key: "clusters".

{{
  "clusters": [
    {{"name": "Name of Cluster 1", "description": "A very information dense description of the cluster", "
    keywords": ["keyword 1", "keyword 2", ...] }},
    {{"name": "Name of Cluster 2", "description": "A very information dense description of the cluster", "
    keywords": ["keyword 3", "keyword 4", ...] }},
    ...
  ],
}}

---

## Your Response
```

Figure 9: Prompt template used for ClusterLLM to be used in top-down tree construction i.e. clustering a given set of keywords into $[M_{min}, M_{max}]$ clusters.

```
You are an expert AI analyst and summarizer. Your mission is to create a highly informative and "
discriminative signpost" for a navigating search agent. This signpost (a summary) must guide the agent to
the correct cluster of nodes to answer a user's query.

You will follow a strict, step-by-step cognitive process. You must analyze the children nodes in a target
parent node (the "Positive Set").

Prompt ID: {prompt_id} (ignore, this is just for watermarking purposes).

## INPUTS

### POSITIVE SET: Information about the target parent node to be summarized

{positive_set_descriptions}
---

## YOUR TASK & OUTPUT FORMAT

Your entire output must be a single, valid JSON object. Inside this JSON, you will follow the 3-step
thinking process outlined below, populating each field as instructed.

### JSON Structure and Instructions:

{{
  "detailed_fingerprints": [
    // For EACH children node in the POSITIVE SET (target parent node), extract a structured object of its
    key, queryable facts.
    {{
      "one_line_summary": "...", // write a very information dense and very concise one-line summary for
      the information contained in this node
      "key_entities": ["..."], // List a very few key entities which is central to this node
      "genre_or_category": ["..."], // List a few key genre / categories this node can be classified into
      "name": "...", // Name the node
    }}
  ],
  "common_theme": "...", // Reason deeply what are the common themes between the nodes in the POSITIVE SET
  "summary": "...", // Based on step 1 and step 2, write a very information dense description of the target
   node, **make sure to include all key entities**.
}}

---

## Your Response
```

Figure 10: Prompt template for generating bottom-up summaries of a group of nodes.

