# OpenReview forum: "LLM-guided Hierarchical Retrieval"
_ICLR.cc/2026/Conference — Submitted to ICLR 2026_

### Official Review · Reviewer_x4dg · 2025-10-31

**Soundness:** 3
**Presentation:** 3
**Contribution:** 2
**Rating:** 4
**Confidence:** 4

**Summary:**

This paper presents LATTICE, a training-free hierarchical retrieval framework designed to help large language models (LLMs) efficiently search large document corpora with logarithmic search complexity. The authors aim to overcome key limitations in current LLM-based information retrieval (IR) systems—namely, the bottlenecks in retrieve-then-rerank pipelines and the scalability challenges of long-context methods.
LATTICE operates in two stages: (1) Offline Stage – Documents are organized into a semantic tree through either a bottom-up or top-down LLM-driven clustering strategy. (2) Online Stage – A search LLM navigates this tree using a traversal algorithm that estimates calibrated latent relevance scores from inherently noisy and context-dependent LLM judgments. These scores are aggregated into a path relevance metric that guides global search decisions.
Empirical results show that LATTICE achieves state-of-the-art zero-shot performance on the reasoning-heavy BRIGHT benchmark, outperforming strong baselines in Recall@100 and nDCG@10, and performing comparably to heavily fine-tuned systems on static corpus subsets.

**Strengths:**

1. The proposed tree search algorithm and offline tree construction methods are sound and empirically validated.
2. The paper provides comprehensive analyses illustrating the advantages and mechanisms of the approach.
3. The writing is clear and well-organized.

**Weaknesses:**

1. The proposed solution appears to perform well only under high online LLM budget settings. As shown in Figure 3, performance drops notably when the budget is low, falling behind the baselines.
2. As shown in Table 3, performance is highly sensitive to the choice of offline tree construction strategy (bottom-up vs. top-down). This suggests that selecting the right strategy requires prior knowledge of the corpus structure, which may not always be available in practice.
3. The experiments are only conducted on the Gemini-2.5 family. It would be helpful to see results on other or smaller open-source models for broader validation.
4. The model comparisons may not be entirely fair, since the proposed method uses Gemini-2.5-flash at all stages, while the baselines rely on GPT-4-based query expansion.
5. The evaluation is limited to the BRIGHT dataset. It would be useful to see how the method performs on other retrieval datasets.
6. Compared to other approaches that perform offline indexing on BRIGHT—such as
*Imagine All The Relevance: Scenario-Profiled Indexing with Knowledge Expansion for Dense Retrieval (Lee et al.)* and
*EnrichIndex: Using LLMs to Enrich Retrieval Indices Offline (Chen et al.)*
—the previously built offline tree structure needs to be reclustered or redivided whenever new documents are added, which could lead to high maintenance costs over time.

**Questions:**

See weakness

---

> ### Author Response · Authors · 2025-11-24
>
> We thank Reviewer x4dg for the positive assessment of our tree search algorithm, comprehensive analyses, and clear writing. We appreciate your recognition that the proposed approach is sound and empirically validated. Please find below our detailed responses to the specific concerns raised.
>
> ### **Performance under Low Budgets (W1)**
>
> We acknowledge that LATTICE is not optimal for low-latency constraints, as the traversal must navigate the tree depth to even reach to a single candidate leaf node. However, we maintain that this computational investment is still justified in following settings:
>
> - Diminishing Returns of Reranking for complex retrieval: As shown in Figure 3 (Right), standard retrieve-then-rerank pipelines show limited performance gains from additional compute. This is because the performance is bounded by the recall of the initial retriever. In contrast, LATTICE can use the additional compute to improve recall by actively searching the corpus. There are many practical scenarios (e.g., Deep Research, Financial QA, Extremely long doc QA) that allow for higher latency and where the accuracy of retrieved results is prioritized over low latency.
>
> - Generating Strong Distilled Training data: One can leverage these accurate search trajectories in offline setting to distill faster, specialized search models or train efficient encoders, optimizing the cost-latency trade-off.
>
> ### **Results with Open-Source Models (W2)**
> To address the concern regarding proprietary models, we evaluated our framework using the Qwen3-VL family of open-weights models (8B, 30B-A3B, and 32B) without changing anything else in the system. The results indicate that the 32B model can closely approximate the performance of Gemini-2.5-Flash, suggesting that the traversal mechanism is effective provided the underlying model possesses sufficient capability. The 8B model performance is lacking while 30B-A3B MoE model with 3B active params gets reasonable performance. We believe that smaller models can get comparable performance with task-specific fine-tuning.
>
> *Table 8: Performance with Open-Source Models (Zero-Shot, nDCG@10)*
>
> | Model | Biology | Economics | Earth Science | TheoremQA theorems
> | :--- | :--- | :--- | :--- | :--- |
> | Qwen3-VL-8B-Instruct | 48.9 | 32.5 | 50.3 |  36.2 |
> | Qwen3-VL-30B-A3B-Instruct | 56.9 | 41.1 | 52.2 | 41.9 |
> | Qwen3-VL-32B-Instruct | 64.0 | 42.4 | 61.1 | 41.8 |
> | Gemini-2.5-Flash (main paper results) | 64.4 | 45.4 | 62.4 | 47.2 |
>
> ### **Fairness of Model Comparison (W3)**
> We respectfully disagree with the statement that comparing with GPT-4 based query expansion is weaker as GPT-4 is generally regarded as a more capable and expensive model than Gemini-2.5-Flash. Previous works (XRR2) utilize GPT4 query expansion because it empirically gives better results than flash. We report results of BM25 with Gemini-2.5-flash vs GPT-4 query expansion on biology subset below to clarify further.
>
> *Table 10: Gemini-2.5-flash vs. GPT-4 Query Expansion (Biology Subset)*
>
> | Method | nDCG@10 | Recall@100 |
> | :--- | :--- | :--- |
> | BM25 with GPT-4 QE | 60.8 | 89.5 |
> | BM25 with Gemini-2.5-flash QE | 59.9 | 85.4 |
>
> ### **Sensitivity of Tree Construction (W4)**
> We agree that the choice of construction strategy (Bottom-Up vs. Top-Down) impacts performance. While we believe most practitioners have sufficient knowledge of their corpus structure (e.g., hierarchical threads vs. independent articles) to select the appropriate method, we agree that a unified, learned construction method is desirable. We draw a parallel to Approximate Nearest Neighbor Search (ANNS) methods - early indexing methods like KD-trees or LSH were data-dependent and required selection based on data type. Modern methods like HNSW are learned specifically for the search task. Currently, our tree construction is unsupervised and oblivious to the search process. We believe the next logical step (which some of our early experiments with dynamic indexing show promise for - please see Table 6 in response to reviewer Qcbe) is to explore semi-supervised index learning, where the tree structure is optimized specifically for LLM navigability, similar to how graph-based indices are optimized for vector search.

---

> > ### Author Response · Authors · 2025-11-24
> >
> > ### **Evaluation on BEIR Datasets (W5)**
> > We prioritized the BRIGHT benchmark because it is the current standard testbed for complex retrieval, where retrieval requires sufficient reasoning to surface right results - allowing a considerable room for improvement in the quality of the retrieval system. Although we believe BRIGHT represents many diverse tasks and is of considerable scale (with 4 subsets of the benchmark having $>150K$ corpus size), we acknowledge reviewer's concerns related to evaluation limited to a particular setting and showing promise to handle million scale corpus. To address this we evaluated LATTICE on three different scale and diverse tasks from the BEIR benchmark: NQ ($2.68M$ passages, answer web search queries), SciDocs ($25K$ passages, find related papers), and SciFact ($5K$ passages, find passages that refute or prove a claim).
> >
> > As shown below, LATTICE performs competitively on these tasks. We would like to emphasize that these benchmark results (specially NQ) should be taken with a grain of salt as (a) Some of these datasets come with the training set which can help disambiguate what is considered correct by the dataset which LATTICE doesn't have access to (b) There is a limited room for improvement in simpler benchmarks. Still, we hope these results helps alleviate concerns regarding scalability and generalizability of LATTICE beyond BRIGHT.
> >
> > *Table 6: Evaluation on BEIR (nDCG@10)*
> >
> > | Dataset | Corpus Size | BGE-large-en-v1.5 | BGE-reranker-large | LATTICE |
> > | :--- | :--- | :--- | :--- | :--- |
> > | SciFact | 5K | 74.6 | 74.1 | 75.8 |
> > | SciDocs | 25K | 22.6 | 17.0 | 22.4 |
> > | NQ | 2.68M | 55.0 | 69.0 | 66.8 |
> >
> >
> > ### **Index Updates (W6)**
> > Regarding the maintenance of the tree in dynamic settings, please see our detailed response and Table 5 to Reviewer Qcbe. In summary, we propose an incremental update strategy where new nodes are inserted via searching them in the existing index and mapping them to the top scoring leaf cluster. Experiments suggest this can actually improve performance by allowing the placement of inserted document to be aligned with the search process. Thus it should be possible to effectively handle moderate updates without requiring full index reconstruction.
> >
> > ---
> >
> > We are happy to provide further clarifications as needed. Thanks for your thoughtful feedback!

---

> > > ### Author Response · Authors · 2025-11-26
> > >
> > > Dear Reviewer, we wanted to check if our rebuttal has sufficiently addressed your concerns or if you have any follow-up questions. Please let us know if there are any critical experiments you would still like to see, so we can ensure they are completed within the remaining discussion period.

---

### Official Review · Reviewer_3E2f · 2025-11-02

**Soundness:** 3
**Presentation:** 3
**Contribution:** 2
**Rating:** 4
**Confidence:** 4

**Summary:**

This paper addresses the challenge of using LLMs for complex information retrieval over large corpora, where existing methods are either limited by embeddings, difficult to update, or computationally infeasible.
The authors propose LATTICE, a hierarchical retrieval framework that structures a document corpus into a semantic tree. An LLM then navigates this tree  to guide the search.
The experiments are employed on the BRIGHT benchmark, demonstrating good improvements in recall and nDCG compared with baselines.

**Strengths:**

1. This paper proposes a good LLM-guided hierarchical information retrieval framework. It designs multiple strategies to construct the original corpus into a hierarchical tree during the offline stage, and carefully designs navigation strategies for the online stage.
2. The paper is  readable, making it easy for readers to understand the authors' motivation and methodology. The complex strategies in the online stage are explained through formulas and pseudocode.
3. Experiments demonstrate that the proposed method achieves better performance under a zero-shot setting compared to baseline models.

**Weaknesses:**

1. The paper lacks obvious innovation, as previously published works have already explored similar ideas[1]: transforming large-scale knowledge corpora into hierarchical trees and designing navigation strategies for traversal and filtering.
2. The paper only employs the BRIGHT benchmark for experiments. Although this benchmark contains multiple subsets, these subsets were all constructed by the same research team, resulting in a uniform pattern across them. This raises concerns about the generalizability of the proposed method.
3. Compared to the fine-tuned DIVER v2, the method proposed in this paper does not demonstrate a clear performance advantage. Although the proposed approach requires no training cost, it incurs offline construction costs. The paper fails to provide a clear comparison between the offline construction cost of their method and the training cost of fine-tuned DIVER v2. This leads to doubts about whether the proposed method achieves only marginal performance gains at a potentially higher overall cost.

[1] Hierarchical Document Refinement for Long-context Retrieval-augmented Generation. ACL 2025

**Questions:**

1. What is the core innovation of the proposed method compared to previous work [1] ?
2. Which one is higher: the offline construction cost of this method or the training cost of fine-tuned DIVER v2?
3. The offline construction phase uses Gemini-2.5-flash. Have other large language models been tried, and how much would using different models impact the final results?

[1] Hierarchical Document Refinement for Long-context Retrieval-augmented Generation. ACL 2025

---

> ### Author Response · Authors · 2025-11-24
>
> We thank Reviewer 3E2f for recognizing our framework as well-designed, readable, and for noting the improvements demonstrated in our zero-shot experiments.
>
> ### **Novelty justification and comparison to LongRefiner (W1, Q1)**
>
> While both works utilize hierarchical structures, we would like to clarify that LATTICE and LongRefiner address different stages of the IR pipeline and operate at vastly different scales.
>
> LongRefiner is a post-retrieval refiner designed to compress context. It operates on a small set of *already retrieved* documents (e.g., top-8), parsing each into a local structural tree (Sections $\to$ Paragraphs) to filter out noise before generation. It cannot find documents missed by the initial retriever. In contrast, LATTICE is a complete IR pipeline parameterized by a single generative LLM model. We construct a global semantic index over the *entire corpus* (100k+ documents). Our tree represents the semantic topology of the full dataset, not the internal structure of a single document. More importantly, LATTICE uses the end LLM itself to actively search the "haystack" from the root to find relevant documents that standard retrievers can miss.
>
> | Feature | LongRefiner | LATTICE (Ours) |
> | :--- | :--- | :--- |
> | **Core Task** | Refinement (Context Compression) | Retrieval + Ranking  |
> | **Goal** | Remove noisy chunks from *retrieved* docs. | Find relevant docs in *entire* corpus. |
> | **Input Scope** | Query + Top-$k$ Retrieved Docs (e.g., 8) | Query + Full Corpus (e.g., 100,000+) |
> | **Tree Structure** | Local Parse Tree (Section $\to$ Para) representing internal document structure. | Global Semantic Tree representing topic clusters across the corpus. |
>
> **Core Contributions**
> To further clarify our novelty, we emphasize three key contributions:
> 1.  **End-to-end LLM-relevance based retrieval**: To the best of our knowledge, LATTICE is one of the first work to use an LLM to directly score and navigate a corpus of $O(100K+)$ documents without relying on a separate, lower-quality retriever to prune the search space first.
> 2.  **Robust Tree Traversal** We propose and validate the design of the tree traversal algorithm that allows an LLM to navigate this global index efficiently and robustly. We introduce critical mechanisms like **Path Relevance Calibration** (comparing nodes against siblings and seen leaves) to stabilize inherently noisy LLM judgments, validating their necessity through extensive ablations (e.g., Table 2).
> 3.  **LLM-Based Top-down tree construction:** We introduce a top-down tree organization method driven entirely by LLM summarization and clustering. To our knowledge, this specific approach of recursively partitioning a flat corpus into a semantic taxonomy using multi-level summaries has not been explored in the literature.
>
>
> ### **Restriction to BRIGHT Benchmark and Generalizability (W2)**
> Our work is focused on BRIGHT because of our focus on complex retrieval tasks where the relevance definition is more nuanced than just semantic similarity or keyword overlap. We acknowledge that the method although not needed for standard retrieval tasks should show promise to claim generalizability - to this end as requested by other reviewers as well, we report results on 3 tasks from BEIR with different scale and characterstics NQ (2.68M corpus), SciDocs (25K corpus) and SciFact (5K corpus). As shown below, LATTICE performs competitively on these tasks. We would like to emphasize that these benchmark results (specially NQ) should be taken with a grain of salt as (a) Some of these datasets come with the training set which can help disambiguate what is considered correct by the dataset which LATTICE doesn't have access to (b) There is a limited room for improvement in simpler benchmarks. We hope these results helps alleviate concerns regarding scalability and generalizability of LATTICE beyond BRIGHT.
>
> *Table 6: Evaluation on BEIR (nDCG@10)*
>
> | Dataset | Corpus Size | BGE-large-en-v1.5 | BGE-reranker-large | LATTICE |
> | :--- | :--- | :--- | :--- | :--- |
> | SciFact | 5K | 74.6 | 74.1 | 75.8 |
> | SciDocs | 25K | 22.6 | 17.0 | 22.4 |
> | NQ | 2.68M | 55.0 | 69.0 | 66.8 |

---

> ### Author Response · Authors · 2025-11-24
>
> ### **Comparison to DIVER-v2 (W3)**
> We would like to note that DIVER-v2 is not a direct comparison for our method as it is an ensemble of four distinct systems: (i) BM25 with Query Expansion, (ii) Diver-Retriever, (iii) a pointwise ranker (Qwen3-32B), and (iv) a listwise ranker (DeepSeek-v3). Notably, without the DeepSeek component (i.e. DIVER-v1), their performance (41.6) is lower than ours (42.0). Moreover, as we show below simply combining our method with BM25 gives a performance boost to our numbers - to avoid confounding the effect of ensembling we report numbers for our method in isolation and that's why we believe the XRR2 method is more controlled and relevant baseline comparison for LATTICE which is a pure retrieve then rerank model using the same reranking model.
>
> *Table 9: LATTICE with BM25*
>
> | Method | Average            | Bio   | Earth | Eco   | Psyc  | Robo  | Stack | Sust  | Leetcode | Pony  | Aops  | TheoQ | TheoT |
> |----------------|-------|-------|-------|-------|-------|-------|-------|----------|-------|-------|-------|-------|----|
> | LATTICE        | 42.0 | 64.4 | 62.3 | 45.3 | **57.3** | 47.5 | 37.5 | 46.3    | **19.8** | 33.9 | **12.0** | 30.1 | 47.8 |
> | LATTICE + BM25 | **43.6** | **69.8**  | **63.8** | **45.3** | 54.0 | **48.1** | **43.1**  | **46.4** | 19.4  | **39.9**  | 11.7 | **32.9** | **48.6** |
>
> **Regarding Training Cost vs. Tree Building Cost (Q2)**
>
> We acknowledge that semantic tree building process is not cheap but can be done in reasonable time - for a 100K corpus it takes ~3 hours when building the tree in bottom-up fashion and ~5-6 hours when building the tree in top-down manner. It is hard to directly compare it with DIVER's training cost as it comes with the cost of accumulating training data and furthermore training a specialized model may hinder the generalization of the underlying model for a new task vs building a semantic tree which is fully unsupervised and generalizable to any set of queries for a given corpus.
>
> ---
>
> We are happy to provide further clarifications as needed. Thanks for your thoughtful feedback!

---

> > ### Author Response · Authors · 2025-11-26
> >
> > Dear Reviewer, we wanted to check if our rebuttal has sufficiently addressed your concerns or if you have any follow-up questions. Please let us know if there are any critical experiments you would still like to see, so we can ensure they are completed within the remaining discussion period.

---

### Official Review · Reviewer_m8ts · 2025-11-02

**Soundness:** 3
**Presentation:** 2
**Contribution:** 2
**Rating:** 4
**Confidence:** 4

**Summary:**

The paper introduces LATTICE, a hierarchical retrieval framework that enables large language models (LLMs) to perform reasoning-driven search over large document corpora with logarithmic complexity. It organizes the corpus into a semantic tree structure using either a bottom-up agglomerative or top-down divisive strategy, and employs an LLM-guided traversal algorithm that estimates calibrated relevance scores to navigate the hierarchy effectively. The framework is training-free and demonstrates strong zero-shot retrieval performance on the reasoning-intensive benchmark BRIGHT.

**Strengths:**

1) The paper presents a hierarchical retrieval framework that integrates LLM reasoning with structured corpus organization.
2) The proposed method is clearly motivated and effectively described.
3) The framework shows potential for improving reasoning-oriented retrieval.

**Weaknesses:**

1) The approach has not been validated on large-scale, open-domain corpora, leaving its scalability and generalization uncertain.
2) The discussion of related work is incomplete, omitting several recent advances in hierarchical and structure-aware retrieval.
3) The paper lacks analysis of efficiency and computational cost during both tree construction and traversal stages.
4) The presentation could be improved. The paper introduces the search process before explaining tree construction and lacks a concluding section summarizing key insights and limitations.
5) The evaluation relies solely on the proprietary Gemini-2.5-flash model and a single reasoning-intensive benchmark, limiting the understanding of model dependence and robustness across different retrieval settings.

**Questions:**

1) How does the semantic hierarchy scale in both construction and traversal time when applied to large-scale, open-domain corpora (e.g., millions of documents)?
2) Beyond BRIGHT, have the authors evaluated LATTICE on more general retrieval datasets such as MS MARCO or Natural Questions to assess robustness and generalization?
3) How do smaller or open-source LLMs perform within this framework? Is the approach dependent on the reasoning strength of proprietary models like Gemini-2.5-flash?
4) Could the authors provide a quantitative or qualitative comparison of the bottom-up vs. top-down semantic tree construction strategies? In what data conditions should one prefer either method?
5) Since traversal efficiency is central to the claim of logarithmic complexity, can the authors provide empirical runtime comparisons against standard reranking pipelines?
6) How is the semantic hierarchy updated when new documents are added? Does the model require full reconstruction, or can it support incremental updates?

---

> ### Author Response · Authors · 2025-11-24
>
> We thank Reviewer m8ts for acknowledging that our framework is clearly motivated and effectively described, and for recognizing its potential for improving reasoning-oriented retrieval.
>
> ### **Validation on Large-Scale Open-Domain Corpora (W1, W5, Q1, Q2)**
>
> Although, we maintain that BRIGHT represents many diverse tasks and is of considerable scale that represents valid setting of retrieval tasks in practical applications (with 4 subsets of the benchmark having $>150K$ corpus size), we acknowledge reviewer's concerns related to evaluation limited to a particular setting and showing promise to handle million scale corpus. To address this we evaluated LATTICE on three different scale and diverse tasks from the BEIR benchmark: NQ ($2.68M$ passages, answer web search queries), SciDocs ($25K$ passages, find related papers), and SciFact ($5K$ passages, find passages that refute or prove a claim).
>
> We hope these results helps alleviate concerns regarding scalability and generalizability of LATTICE beyond BRIGHT. We would also like to emphasize that these benchmark results (specially NQ) should be taken with a grain of salt as (a) Some of these datasets come with the training set which can help disambiguate what is considered correct by the dataset which LATTICE doesn't have access to (b) There is a limited room for improvement in simpler benchmarks.
>
> *Table 6: Evaluation on BEIR (nDCG@10)*
>
> | Dataset | Corpus Size | BGE-large-en-v1.5 | BGE-reranker-large | LATTICE |
> | :--- | :--- | :--- | :--- | :--- |
> | SciFact | 5K | 74.6 | 74.1 | 75.8 |
> | SciDocs | 25K | 22.6 | 17.0 | 22.4 |
> | NQ | 2.68M | 55.0 | 69.0 | 66.8 |
>
> ### **Efficiency and Computational Cost Analysis (W3, Q5)**
>
> Below we provide the breakdown of online traversal latency on the Biology subset of BRIGHT. We report metrics for both the API-based Gemini-2.5-Flash and locally hosted (on 4xA100 80GB setup) Qwen3 models.
>
> *Table 7: Empirical runtime analysis with API based calls and locally run LLMs*
>
> Model | Throughput (num parallel search steps / second)  | Latency (seconds / single search step) |
> | :--- | :--- | :--- |
> | Gemini-2.5-Flash with thinking (API) | - | 10s |
> | Gemini-2.5-Flash without thinking (API) | - | 0.8s |
> | Qwen3-VL-8B-Instruct (Local) | 6.3 steps/s | 1.0s |
> | Qwen3-VL-30B-A3B-Instruct (Local) | 5.0 steps/s | 1.8s |
> | Qwen3-VL-32B-Instruct (Local) | 1.8 steps/s | 4.6s |
>
> Note that in our standard setup for a query we run total $2$ (beam-size) $\times$ $20$ (iterations) $ = 40$ search steps per query, the 2 parallel steps in a beam can be effectively parallelized but iterations need to happen sequentially hence for a query one can expect ~16s end-to-end latency when using Gemini-2.5-flash API without thinking and ~20s when using an 8B local Qwen model though there is a big efficiency gain in parallelization when doing batch inference as suggested by the throughput numbers.
>
> We would also like to refer the reviewer to Figure 3 in the main paper which analyzes the trade-off between performance (nDCG@10) and computational cost (measured in input tokens to the LLM). This comparison benchmarks LATTICE against the retrieve-then-rerank baselines, showing that LATTICE yields better performance per token once a minimum traversal depth is reached.
>
> Tree construction for a 100K corpus takes ~3 hours when building the tree in bottom-up fashion and ~5 hours when building the tree in top-down manner.
>
> ### **Performance of Open-Source LLMs to assess model dependence (W5, Q3)**
>
> We evaluated our framework using the Qwen3-VL family of open-weights models (8B, 30B-A3B, and 32B) without changing anything else in the system. The results indicate that the 32B model can closely approximate the performance of Gemini-2.5-Flash, suggesting that the traversal mechanism is effective provided the underlying model possesses sufficient capability. The 8B model performance is poor while 30B-A3B MoE model with 3B active params gets reasonable performance while being significantly faster. We believe that smaller models may require task-specific fine-tuning to get comparable performance.
>
> *Table 8: Performance with Open-Source Models (Zero-Shot, nDCG@10)*
>
> | Model | Biology | Economics | Earth Science | TheoremQA theorems
> | :--- | :--- | :--- | :--- | :--- |
> | Qwen3-VL-8B-Instruct | 48.9 | 32.5 | 50.3 |  36.2 |
> | Qwen3-VL-30B-A3B-Instruct | 56.9 | 41.1 | 52.2 | 41.9 |
> | Qwen3-VL-32B-Instruct | 64.0 | 42.4 | 61.1 | 41.8
> | Gemini-2.5-Flash (main paper results) | 64.4 | 45.4 | 62.4 | 47.2 |

---

> ### Author Response · Authors · 2025-11-24
>
> ### **Bottom-Up vs. Top-Down Construction Strategy (Q4)**
>
> Bottom-up construction is useful when there is already some level of hierarchical organization in the corpus for e.g. if the passages in the corpus belong to a bigger document then we can impose this structure when building the bottom-up tree. Because it relies on embedding based clustering when a prior hierarchy is not known - it struggles with building clean thematic clusters required for creating a taxonomy of topics - an example is that the embedding of a cluster about "tv dramas of crime genre" is consistently embedded closer to "law & criminal justice" than a cluster of "film and entertainment". In contrast, the top down approach can first extract a fuzzy taxonomy using hierarchical keyword generation and then an LLM can create clean and consistent themes. The downside is that it treats every document to be independent. We believe one can use a mix of both of thes approaches to build a better hierarchy by using the bottom-up summaries where the hierarchy structure is available and use the top-down approach on top of the resulting clusters.
>
> ### **Updating the Hierarchy (Q6)**
>
> Please see our detailed response (Table 5) to Reviewer Qcbe. In summary, we propose an incremental update strategy where new nodes are inserted via searching them in the existing index and mapping them to the top scoring leaf cluster. Experiments suggest this can actually improve performance by allowing the placement of inserted document to be aligned with the search process. Thus we believe it should be possible to effectively handle moderate updates without requiring full index reconstruction.
>
> ### **Writing Changes(W2, W4)**
>
> 1. Related Work Coverage: while we compared LATTICE against vector hierarchies (HNSW) and summary-trees (RAPTOR), we will expand Section 2 to include LongRefiner and EnrichIndex (suggested by other reviewers). We welcome any specific references the reviewer recommends and will ensure they are integrated.
>
> 2. Regarding the ordering of Search (Section 3.2) prior to Tree Construction (Section 3.3): our rationale was to first define the inference mechanism to motivate the specific requirements for the tree's node summaries. We recognize this may hinder readability, we will reorganize this section as suggested to improve the logical flow.
>
> 3. Conclusion and Limitations: due to space constraints, the Limitations and Future Work discussion was placed in Appendix A. In the final version, we will reintegrate a formal Conclusion section into the main body, summarizing the key insights and explicitly discussing the limitations regarding dynamic corpora and latency.
>
> ---
>
> We are happy to provide further clarifications as needed. Thanks for your thoughtful feedback!

---

> > ### Author Response · Authors · 2025-11-26
> >
> > Dear Reviewer, we wanted to check if our rebuttal has sufficiently addressed your concerns or if you have any follow-up questions. Please let us know if there are any critical experiments you would still like to see, so we can ensure they are completed within the remaining discussion period.

---

### Official Review · Reviewer_Qcbe · 2025-11-10

**Soundness:** 3
**Presentation:** 3
**Contribution:** 3
**Rating:** 6
**Confidence:** 4

**Summary:**

The paper proposes LATTICE, an LLM-guided hierarchical retrieval framework that organizes a corpus into a semantic tree offline and lets a “search LLM” traverse it online using calibrated path-relevance scores. On BRIGHT, it achieves strong zero-shot results and favorable cost–quality scaling compared to retrieve-then-rerank, with clear ablations on calibration and traversal design.

**Strengths:**

1. The proposed method is well-motivated and novel. The latent-score calibration and path-relevance update are interesting and reasonable.

2. Strong empirical results and thoughtful analysis.

3. Under larger token budgets, the method scales better than reranking.

**Weaknesses:**

1. Offline tree construction is expensive and appears data-sensitive (e.g., Table 3). Maintaining the tree for dynamic corpus (add/edit/delete) is nontrivial, as internal summaries can become stale. These issues may hinder real-world adoption.

2. This paper could benefit from more comparisons to agentic methods. The argument that “agents call a retrieval tool while LATTICE is the core retrieval mechanism” is not fully convincing. Both approaches rely on text embeddings but mainly in that LATTICE pre-clusters and has the LLM walk over clustered tree anchors, whereas an agent can pick an anchor (the query embedding) and check neighboring documents.  I believe more in-depth comparisons would help.

3. It is unclear how corpus size affects performance (both tree construction and search under a given budget). The BRIGHT corpus is relatively artificial and small, comparisons on larger datasets (eg, BEIR) would be helpful.

**Questions:**

1. How LATTICE maintains the diversity of retrieved results

---

> ### Author Response · Authors · 2025-11-24
>
> We sincerely thank Reviewer Qcbe for the positive feedback on our method's novelty, strong empirical results, and analysis. Please find below our detailed responses to the specific concerns raised.
>
> ### **Tree Updatability (W1)**
> To address the limitation of updatability of the semantic tree, we explore a search and insert update. More specifically, we update the search index with new documents by traversing the existing tree using the LATTICE's search algorithm to identify the optimal (one that scores the highest) leaf cluster for insertion. We validate this approach on one of the subset of BRIGHT by withholding all the gold documents for the eval queries, constructing the tree with the remaining documents, and then inserting the withheld documents via traversal. The final performance with the dynamically inserted tree actually improved compared to the tree built using full corpus (as in our original experiments). This suggests that aligning the document placement with the search logic - analogous to graph construction of ANNS methods like HNSW is a viable update mechanism. While running these experiments we also realized that this can be a promising direction for robustly constructing the full semantic tree i.e. starting with the initial tree and incrementally do search-then-insert each document to its best path found.
>
> *Table 5: Dynamic Insertion Experiment (TheoremQA)*
>
> | Method | nDCG@10 | Recall@100 |
> | :--- | :--- | :--- |
> | Static Construction (full corpus, original) | 47.4 | 73.9 |
> | Dynamic Insertion (insert gold documents later) | 49.7 | 81.1 |
>
> Regarding the staleness of internal summaries, we believe as long as the search index contains a valid path (one that gets a high score) for the new document to be inserted - we can skip updating the internal summaries but in case of massive updates we acknowledge there is a need for updating the summaries. One possible strategy could be to re-cluster affected internal nodes based on the severity of the update (e.g. number of new documents inserted in the subtree) but it needs to be carefully designed to avoid catastrophic changes to the tree structure. We hope future work can explore dynamic tree maintenance in more detail.
>
> ### **Comparison to Agentic Methods (W2)**
> This is an interesting and valid interpretation of agentic methods that they can choose to query a local neighborhood (defined using embeddings) in the corpus based on what query they ask. LATTICE also explores different neighborhoods in the corpus defined by the semantic tree but we would like to point out a few distinctions:
>
> - Global structural awareness vs. unconstrained querying: standard agents operate in an unconstrained action space, generating queries to probe an opaque index. LATTICE allows the LLM to get a global view of the corpus and condition its decisions of what to explore based on what is available in the corpus which is desirable when the prior knowledge about what to look for is not clear or murky - can arise in complex retrieval setup.
>
> - Semantic vs. vector topology: as observed in Table 3, defining neighborhoods via embedding proximity (Bottom-Up) is not always optimal. LATTICE's Top-Down approach which utilizes LLM reasoning to induce a coherent semantic taxonomy can be better in certain cases. Moreover, In domains such as large codebases, user manuals, legal documents, etc a hierarchical structure is intrinsic. LATTICE can naturally exploit this native topology, whereas standard agents typically flatten these structures into chunks for dense retrieval.
>
> We believe agentic methods like search-r1 are powerful formulations when the model has a good understanding about what to look for and our approach is similar to these agentic systems but one that allows the llm to interact with the corpus at multiple granularities than just local neighborhoods defined in the embedding space. We will elaborate on this connection in the final manuscript.

---

> > ### Author Response · Authors · 2025-11-24
> >
> > ### **Evaluation Scope and Scalability (W3)**
> > We prioritized the BRIGHT benchmark because it is the current standard testbed for complex retrieval, where retrieval requires sufficient reasoning to surface right results - allowing a considerable room for improvement in the quality of the retrieval system. Although we believe BRIGHT represents many diverse tasks and is of considerable scale (with 4 subsets of the benchmark having $>150K$ corpus size), we acknowledge reviewer's concerns related to evaluation limited to a particular setting and showing promise to handle million scale corpus. To address this we evaluated LATTICE on three different scale and diverse tasks from the BEIR benchmark: NQ ($2.68M$ passages, answer web search queries), SciDocs ($25K$ passages, find related papers), and SciFact ($5K$ passages, find passages that refute or prove a claim).
> >
> > As shown below, LATTICE performs competitively on these tasks. We would like to emphasize that these benchmark results (specially NQ) should be taken with a grain of salt as (a) Some of these datasets come with the training set which can help disambiguate what is considered correct by the dataset which LATTICE doesn't have access to (b) There is a limited room for improvement in simpler benchmarks. Still, we hope these results helps alleviate concerns regarding scalability and generalizability of LATTICE beyond BRIGHT.
> >
> > *Table 6: Evaluation on BEIR (nDCG@10)*
> >
> > | Dataset | Corpus Size | BGE-large-en-v1.5 | BGE-reranker-large | LATTICE |
> > | :--- | :--- | :--- | :--- | :--- |
> > | SciFact | 5K | 74.6 | 74.1 | 75.8 |
> > | SciDocs | 25K | 22.6 | 17.0 | 22.4 |
> > | NQ | 2.68M | 55.0 | 69.0 | 66.8 |
> >
> > ### **Maintaining Diversity (Q1)**
> > While our primary objective was relevance, we believe there are 2 natural places to incorporate diversity in the proposed traversal algorithm (i) modifying the scoring prompt (Figure 7 ) such that the LLM is instructed to penalize redundancy within the candidate slate when scoring (ii) increasing the beam-size during exploration so that the traversal naturally explores parallel ways of answering the query. Based on our qualitative analysis, we observe that LATTICE usually does retrieve diverse perspectives on a query, for e.g. for the query "show me the meaning of being lonely", the model explores both "Music" cluster (for the interpretation a song by blackstreet boys) and "Pscychology" cluster (for the interpretation psychological aspects of loneliness).
> >
> > ---
> >
> > We are happy to provide further clarifications as needed. Thanks for the great points raised!

---

> > > ### Author Response · Authors · 2025-11-26
> > >
> > > Dear Reviewer, we wanted to check if our rebuttal has sufficiently addressed your concerns or if you have any follow-up questions. Please let us know if there are any critical experiments you would still like to see, so we can ensure they are completed within the remaining discussion period.

---

### Author Response · Authors · 2025-12-03
**Final remarks by Authors**

We thank everyone involved in the reviewing process to take their time to provide feedback to our work. As we conclude, we would like to offer a concise summary of our positioning and the key concerns and our response to them.

**What are we proposing and why is it relevant**: this work proposes a search framework where offline compute is spent to build an LLM navigable search index and the LLM is used to navigate the search index at test time using its internal reasoning and understanding of the context. To the best of our knowledge, our work is one of the first to show feasibility and strong results for such a framework at the full scale of retrieval from a large corpus (~100K for BRIGHT and ~2.6M for NQ results added during rebuttal). It shows promise of generative LLMs to perform end-to-end retrieval by themselves i.e. without relying on external search tools. Currently this work can have applications in two scenarios: 1) deep search scenarios i.e. complex tasks where high accuracy and reasoning depth are prioritized over sub-second latency; 2) using it in an offline setting to generate distillation data for training faster and accurate search models.

**Novelty**:
- We propose methods to build an LLM navigable search index in an unsupervised setting. Although the bottom-up tree construction is similar to existing techniques in the literature (such as RAPTOR), the top-down approach which uses the LLM itself to organize the corpus at multiple levels of granularity is novel to the best of our knowledge and is shown to perform better when dealing with independent documents in the corpus.
- We identify challenges with LLM based navigation over a semantic search index i.e. noisy and context dependent judgements and propose a robust navigation algorithm that explores nodes based on path-based scores (instead of local scores) and adds cross branch comparisons to calibrate LLM judgements.

**Comparison to existing methods**:
- Standard retrieve-then-rerank methods hit a performance ceiling because they are bounded by the initial recall of embedding-based retrieval. In contrast, our results show that LATTICE breaks this ceiling - scaling more effectively with test-time compute (Figure 3).
- Current agentic systems operate in an unconstrained action space, generating queries to probe an opaque search tool - essentially guessing keywords to find a local neighborhood of documents, LATTICE allows the LLM to get a global view of the corpus and condition its decisions of what to explore based on what is available in the corpus which is desirable when the prior knowledge about what to look for is not clear.

**Generalization to large-scale and open-source settings**: To address concerns about relying solely on the BRIGHT benchmark and proprietary models:

- **Beyond BRIGHT**: In Table 6, we expanded evaluation to the BEIR benchmark with upto 2.68M sized corpus (NQ) to test robustness and scalability. LATTICE performs competitively to strong retrieve-then-rank baselines optimized for such benchmarks, showing it handles standard open-domain tasks well. However, we maintain that reasoning-intensive benchmarks like BRIGHT (where keyword based search fails) are more appropriate to evaluate our proposed method.

- **Open weights**: In Table 8, we show that LATTICE's performance is not locked to proprietary models. The open-source Qwen3-VL-32B achieves performance comparable to Gemini-2.5-Flash supporting that the framework is robust, provided the underlying LLM has sufficient reasoning capability.

**Dynamic updates and maintenance of semantic tree**: In Table 5, we addressed the concern regarding the cost of maintaining the semantic tree (Qcbe, x4dg). Our new experiments with a search-and-insert strategy demonstrate that we can update the index incrementally without full reconstruction. Interestingly, inserting new documents by having the LLM search for their best location in the tree actually improved performance compared to a static build, suggesting a path toward self-optimizing indices.

**More details around cost analysis**: In table 7, we have provided both throughput and latency of online traversal using different open (run in a local setup) and proprietary (API-based) models. We have also added average runtimes for building semantic tree in bottom-up and top-down setting for BRIGHT datasets.

---

We hope our rebuttal response has sufficiently answered reviewer's concerns. We would like to thank again to the reviewers for their feedback, it has been very helpful for us to position our work better, improve our evaluation and gain confidence in our work's scalability and generalizability.

Best,
Authors of submission #21440

---

### Meta-Review · Area_Chair_SqCg · 2026-01-07

**Summary:**

This paper introduces LATTICE, a training-free hierarchical retrieval framework that organizes a corpus into a semantic tree offline and enables an LLM to navigate it online using calibrated path-relevance scores, aiming to outperform retrieve-then-rerank pipelines under higher reasoning and token budgets. Reviewers generally agreed the method is well-motivated, clearly described, and empirically strong on BRIGHT, with careful ablations and analyses supporting its design choices. Major concerns focused on scalability, dynamic corpus updates, cost and latency, fairness of comparisons, and generalization beyond BRIGHT. In response, the authors added BEIR evaluations up to million-scale corpora, runtime and cost analyses, open-source model results, incremental update experiments, and clearer positioning relative to agentic and prior hierarchical methods.

**Reviewer Concerns:**

Reviewer Qcbe’s concerns about tree updatability, comparison with agentic methods, scalability to larger corpora, and diversity were largely addressed through dynamic insertion experiments, BEIR evaluations, and conceptual clarifications, though long-term maintenance costs remain partly open.

Reviewer m8ts’s concerns regarding scalability, efficiency, model dependence, evaluation scope, construction strategy comparison, and hierarchy updates were comprehensively addressed with BEIR results, detailed runtime tables, open-source model experiments, and explicit bottom-up vs top-down guidance.

Reviewer 3E2f’s questions about novelty relative to prior hierarchical work, limited benchmarks, and cost comparison with DIVER v2 were addressed through a detailed conceptual comparison, added BEIR results, and explicit discussion of offline construction cost versus training cost, though novelty skepticism may persist at a subjective level.

Reviewer x4dg’s concerns about low-budget performance, sensitivity to construction strategy, reliance on Gemini models, fairness of comparisons, dataset diversity, and index maintenance were mostly addressed via new analyses, open-source model results, BEIR evaluations, and incremental update strategies, while the low-budget weakness is acknowledged rather than eliminated.

**Reviewer Scores:**

Reviewer Qcbe did not explicitly state a score change, but since most concerns were addressed with new experiments and analyses, their remaining reservations are likely limited to practical deployment complexity.

Reviewer m8ts did not indicate a score update, but the rebuttal appears to have fully addressed all substantive technical questions they raised.

Reviewer 3E2f (4) did not state a score update, and while factual concerns were answered, their skepticism about innovation relative to prior work may remain partially unresolved.

Reviewer x4dg (4) did not state a score update, and although most issues were addressed, the acknowledged trade-off under low-budget settings suggests their concerns are only partially resolved.

---

### Decision · Program_Chairs · 2026-01-26

Reject